# Learning local equivariant representations for quantum operators

**Zhanghao Zhouyin**[2,3]  **Zixi Gan**[2,4]  **Shishir K. Pandey**[5]  **Linfeng Zhang**[2,6]  **Qiangqiang Gu**[1,2,7] [*]

[1]School of Artificial Intelligence and Data Science, University of Science and Technology of China, Hefei, China

[2]AI for Science Institute, Beijing, China

[3]Department of Physics, McGill University, Montreal, Canada

[4]Department of Chemistry, Zhejiang University, Hangzhou, China

[5]Birla Institute of Technology & Science, Pilani-Dubai Campus, Dubai, UAE

[6]DP Technology, Beijing, China

[7]Suzhou Institute for Advanced Research, University of Science and Technology of China, Suzhou, China

## Abstract

Predicting quantum operator matrices such as Hamiltonian, overlap, and density matrices in the density functional theory (DFT) framework is crucial for material science. Current methods often focus on individual operators and struggle with efficiency and scalability for large systems. Here we introduce a novel deep learning model, SLEM (strictly localized equivariant message-passing), for predicting multiple quantum operators that achieves state-of-the-art accuracy while dramatically improving computational efficiency. SLEM's key innovation is its strict locality-based design for equivariant representations of quantum tensors while preserving physical symmetries. This enables complex many-body dependency without expanding the effective receptive field, leading to superior data efficiency and transferability. Using an innovative SO(2) convolution and invariant overlap parameterization, SLEM reduces the computational complexity of high-order tensor products and is, therefore, capable of handling systems requiring the $f$ and $g$ orbitals in their basis sets. We demonstrate SLEM's capabilities across diverse 2D and 3D materials, achieving high accuracy even with limited training data. SLEM's design facilitates efficient parallelization, potentially extending DFT simulations to systems with device-level sizes, opening new possibilities for large-scale quantum simulations and high-throughput materials discovery.

## 1 Introduction

Quantum operators, representing observables and the evolution of quantum systems, are the cornerstone of describing the microscopic world. In modern quantum science, the advent of density functional theory (DFT) (Hohenberg & Kohn (1964); Kohn & Sham (1965)) has elevated single-particle quantum operators, such as the Kohn-Sham Hamiltonian, density matrix, and overlap matrix, to paramount importance in solving complex problems (Jones (2015)). These operators play a crucial role in unravelling electronic structures, predicting material properties, and advancing quantum technologies. However, as we tackle larger and more complex systems, these fundamental operators' efficient and accurate representation has emerged as a pressing challenge in computation, demanding new avenues for innovative methodologies.

Recent advances have incorporated machine learning (ML) techniques to accelerate DFT calculations by directly predicting DFT's output of quantum operators, including charge density (Unke et al. (2021)), overlapping matrix (Yu et al. (2023); Unke et al. (2021)), self-energy (Dong et al. (2024)), wave function (Unke et al. (2021)), and Hamiltonian matrix (Yin et al. (2024); Yu et al. (2023); Gong et al. (2023); Nigam et al. (2022); Unke et al. (2021); Zhong et al. (2023)). By circumventing self-consistent DFT calculations, such methods have the potential to scale up the electronic

---

[*]guqq@ustc.edu.cn

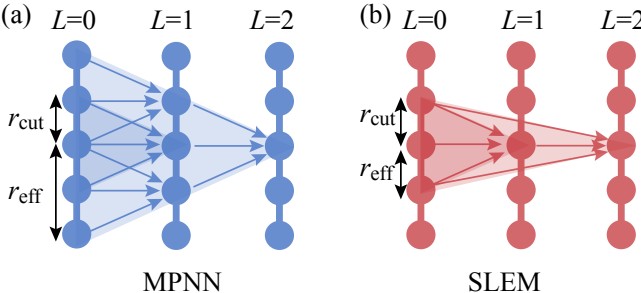

Figure 1: Local design of SLEM vs MPNN.(a) MPNN aggregation. (b) SLEM aggregation. Balls: nodes, sticks: edges, arrows: aggregation. $r_{\text{cut}}$: fixed cutoff; $r_{\text{eff}}$: effective cutoff. $L$: layer index.

structure calculations. Some of the approaches utilize Gaussian regressions (Nigam et al. (2022)), kernel-based models (Nigam et al. (2022)), and neural networks (Li et al. (2022)) to predict invariant Hamiltonian matrix blocks on localized frames. Notably, powerful equivariant message-passing neural networks (E-MPNNs) have demonstrated remarkable accuracy (Han et al. (2024); Musaelian et al. (2023); Batatia et al. (2022); Joshi et al. (2023); Liao & Smidt; Liao et al.; Simeon & De Fabritiis (2024); Passaro & Zitnick (2023); Zitnick et al. (2022)). These networks ensure the output tensor blocks' equivariance, respecting atomic systems' physical priors. Typically, they use iterative updates to build many-body interactions, achieving high accuracy while enlarging the receptive field. This limits parallelization and, consequently, the model's scalability. The storage-intensive quantum tensor prediction tasks exacerbate these limitations, posing considerable challenges for training such models on large datasets or predicting quantum operators for extensive atomic structures . Fortunately, the electrostatic screening counteracts the long-term dependency in a lot of material systems (Huckel & Debye (1923); Resta (1977); Ninno et al. (2006)). Therefore, quantum operators can be decomposed into elements dependent locally on atomic structures, which natrually prefers a strictly local model that avoids expanding the receptive field. The Allegro model (Musaelian et al. (2023)) applied this concept to build ML interatomic potentials (MLIPs), achieving high accuracy and parallelizability. While MLIPs only concern scalars (energy) and vectors (forces) (angular momentum $l = 0$ and $1$) on each node, predicting quantum operators necessitates targeting both node and edge features on high-order spherical tensors (even up to $l = 6$ or $8$). This requires locality and representability for both node and edge. Another significant challenge is computational complexity. To equivariantly mix the features of different angular momentum $l$, tensor products that scale as $\mathcal{O}(l^6)$ are required (Passaro & Zitnick (2023)). This makes model training, especially on heavy atoms, extremely slow. This limitation hinders the development of a unified ML DFT model that generalizes across the periodic table.

This work presents a novel method, the strictly localized equivariant message-passing model (SLEM), for efficient representations of quantum operators. SLEM employs a fully localized scheme to construct high-order node and edge equivariant features for a general representation of quantum operators, including the Hamiltonian and density matrix. As illustrated in Fig. 1, the model embeds localized edge hidden states and utilize them to construct localized node and edge features without including distant neighbours beyond a fixed cutoff range $r_{\text{cut}}$. This design enables the SLEM model to generalize better, parallelize easier, and scale to larger systems. Additionally, a fast and efficient SO(2) convolution (Passaro & Zitnick (2023)) is implemented to reduce the $\mathcal{O}(l^6)$ complexity, with edge-specific training weights, thereby further enhancing the model's accuracy. As for the overlap matrix, it is typically required for property calculations. Previous works have used another E-MPNN that doubles the network sizes (Zhong et al. (2023)) or extracts it from DFT calculations (Gong et al. (2023); Yu et al. (2023)), incurring additional costs or incorporating out-of-loop computation steps that complicate inference. In contrast, we utilized the two-centre integrals and parameterized the overlap matrix with spherical independent scalars (i.e., Slater-Koster (SK) parameters (Slater & Koster (1954))). This method, inspired by the work of DeePTB (Gu et al. (2024)), fits the overlap operator representation with minimal additional cost.

## 2 RELATED WORKS

**Message-passing Neural Networks**   Message-passing neural networks (MPNNs) (Gilmer et al. (2017)) have been widely applied in the modelling of atomic systems due to their exceptional ac-

curacy in capturing the intricate relationships between atomic environments and physical properties (Schütt et al. (2020)). Previous works have predominantly utilized this scheme, achieving remarkably high precision in reported systems (Schütt et al. (2018); Satorras et al. (2021); Han et al. (2024)). However, as the MPNNs updates, the effective cutoff radius ($r_{\text{eff}} = N \times r_{\text{cut}}$) for each atom's features grows linearly with the number of update steps ($N$), as shown in Fig. 1. Consequently, the effective neighbour list scales cubically with $r_{\text{eff}}$, making parallelization intractable. Allegro (Musaelian et al. (2023)) achieved locality by modifying the updating rules by incorporating a hidden pair state that depends partially on the center atom. While this framework works successfully for scalars and $l$=1 vectors in potential energy prediction, it requires further improvement for fitting more general edge and node features. Other local methods avoid iterative updates that enlarge the receptive field, instead creating manually designed descriptors of the local environment (Behler & Parrinello (2007); Bartók et al. (2010); Thompson et al. (2015); Zhang et al. (2018a); Wang et al. (2018); Zhang et al. (2018b)). These methods are generally local, but often balance locality and representation capacity.

**Equivariant Message Passing**   Physical quantities, under the law of nature, should be invariant or equivariant under the spatial and temporal symmetry operations. To model such quantities, a set of neural network models has been developed utilizing equivariant operations. (Thomas et al. (2018); Weiler et al. (2018); Kondor et al. (2018); Kondor (2018)) These neural networks possess physical priors to ensure outputs transform in sync with inputs, making them more generalizable, accurate, and data-efficient in predicting physical quantities. Formally, an equivariant operation from vector space $X$ to $Y$ is defined such that:

$$f(D_X[g]\mathbf{x}) = D_Y[g]f(\mathbf{x}) \quad \forall g \in G, \forall \mathbf{x} \in X$$

where $D_X[g] \in GL(X)$ is the representation of group element $g$ on vector space $X$. Here we consider $O(3)$ group, then $\mathbf{x}, \mathbf{y}$ can be composed by irreducible representation (irreps for short) that are the spherical tensors with angular and magnetic momentum index $l, m$, and parity $p$ such that $|m| \leq l$. Irreps with the same $l$ support addition/subtraction, while a generalized multiplication is defined as tensor product ($\otimes$):

$$(\mathbf{x} \otimes \mathbf{y})_{m_3}^{l_3} = \sum_{m_1, m_2} C_{(l_1, m_1)(l_2, m_2)}^{(l_3, m_3)} \mathbf{x}_{m_1}^{l_1} \mathbf{y}_{m_2}^{l_2}$$

Where $C_{(l_1, m_2)(l_2, m_2)}^{(l_3, m_3)}$ are Clebsch-Gordan (CG) coefficients. Conventional tensor product has the time and memory scales of $O(l_{\max}^6)$ (Passaro & Zitnick (2023)) where $l_{\max}$ is the maximum angular momentum in $\mathbf{x}$ and $\mathbf{y}$. Such complexity poses great challenges for quantum tensor prediction. For example, constructing blocks of $f$-$f$ and $g$-$g$ orbital pairs require irreps of maximum order $l = 6$ and $l = 8$. Such high costs make training for large-size systems nearly impossible.

## 3 MODEL ARCHITECURE

### 3.1 PARAMETERIZE EQUIVARIANT QUANTUM OPERATORS

The equivariant parameterization of quantum operators $\hat{O}$, such as the Hamiltonian and density matrix in the LCAO-based DFT framework, is illustrated in Fig. 2. The matrix element of operator $\hat{O}$ can be expressed as:

$$O_{l_1, l_2, m_1, m_2}^{i,j} = \langle i, l_1, m_1 | \hat{O} | j, l_2, m_2 \rangle \tag{1}$$

Here $i$ and $j$ denote atomic sites, while the angular and magnetic momentum index $l, m$ label the atomic orbitals of the site. We apply the Wigner-Eckart theorem to decomposes the operator indexed by $l_1, l_2$ into a single index $l_3$ that satisfies $|l_1 - l_2| \leq l_3 \leq (l_1 + l_2)$:

$$o_{l_3, m_3}^{i,j} = \sum_{l_1, m_1, l_2, m_2} C_{(l_1, m_1)(l_2, m_2)}^{(l_3, m_3)} O_{l_1, l_2, m_1, m_2}^{i,j} \tag{2}$$

Here, the edge ($i \neq j$) and node ($i = j$) features $o_{l_3, m_3}^{i,j}$ are grouped by the index $m$ into vectors of $\mathbf{o}_{c,l}^{i,j}$ with $c$ accounting for multiple tensors for the same $l$. These features can be computed for hopping ($i \neq j$) and onsite ($i = j$) elements of the quantum operators. Further, by leveraging the Hermitian nature of quantum operators, the parameterized elements can be reduced to upper

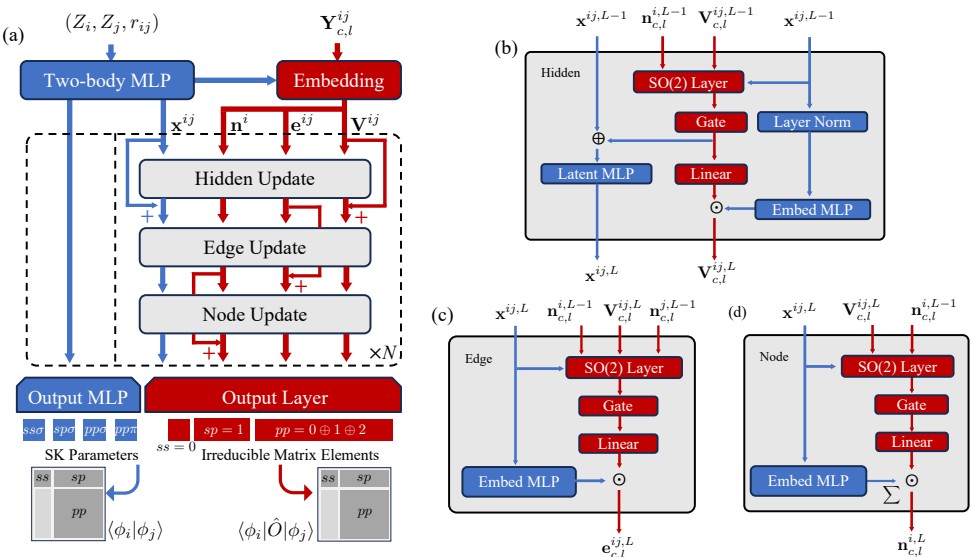

Figure 2: Design of the SLEM model. (a) Hierarchical structure of the model. Starts with the atomic number $Z_i$, the radial and spherical part of the shift vector $r_{ij}$ and $\mathbf{Y}_{c,l}^{ij}$, the initialized hidden features $\mathbf{x}^{ij}$, $\mathbf{V}^{ij}$, along with edge and node features $\mathbf{e}^{ij}$, $\mathbf{n}^i$, are generated. The two-body hidden features predict the SK parameters constructing (off-)diagonal blocks for the overlap operator. Others features are then iteratively updated using the designed strictly localized updating scheme. (b)-(d) shows the hidden update (b), edge update (c), and node update (d). Node and edge features are used to construct the diagonal blocks for quantum operators.

triangular blocks. This is almost half the demanding storage. Then, we standardised $\mathbf{o}_{c,l}^{i,j}$ to balance the variance. Formally:

$$\mathbf{o}_{c,l}^{i,j} = \sigma_{c,l}^{Z_i,Z_j} \hat{\mathbf{o}}_{c,l}^{i,j} + \mu_{c,l}^{Z_i,Z_j} \delta_{l,0} \tag{3}$$

Here, $Z_i$, $Z_j$ are the atom species of atom $i$ and $j$, $\sigma_{c,l}^{Z_i,Z_j}$ and $\mu_{c,l}^{Z_i,Z_j}$ are norm and bias value for each atom and atom-pair. These values are derived from the dataset statistics and applied as weights and biases in an atom/bond type-specific scaling layer. This step helps to resolve the unbalanced norms across diagonal and off-diagonal elements of quantum operators while facilitating using a ReLU-activated network to learn the normalized radial dependent decaying function from 1 to 0.

After supervising on the normalized features $\hat{\mathbf{o}}_{c,l}^{i,j}$, inverse transform from Eq. 2 is applied to reconstruct the predicted operator representations such as Hamiltonian and density matrix blocks:

$$O_{l_1,l_2,m_1,m_2}^{i,j} = \sum_{l_3,m_3} C_{(l_1,m_1)(l_2,m_2)}^{(l_3,m_3)} o_{l_3,m_3}^{i,j} \tag{4}$$

The whole procedure satisfies the rotational, translational and reflectional symmetry.

## 3.2 PARAMETERIZE OF INVARIANT OVERLAP OPERATORS

The overlap matrix, analogous to other quantum operators, is defined as:

$$S_{l_1,l_2,m_1,m_2}^{i,j} = \langle i, l_1, m_1 | j, l_2, m_2 \rangle = \int \phi_{m_1}^{i,l_1}(\mathbf{r}) \phi_{m_2}^{j,l_2}(\mathbf{r} - \mathbf{r}_{ij}) d\mathbf{r} \tag{5}$$

where $\phi_m^{i,l}$ is the orbital from LCAO bases. The overlap matrix also satisfies the equivariance relation. Since the equivariant dependency comes fully from the bases, it is possible to rotate them to align with the axis of $\mathbf{r}_{ij}$, reducing the angular dependence. Therefore, we can further simplify the matrix elements into scalars via the relation (Podolskiy & Vogl (2004)):

$$\int \phi_{m_1}^{i,l_1}(\mathbf{r}) \phi_{m_2}^{j,l_2}(\mathbf{r} - \mathbf{r}_{ij}) d\mathbf{r} = s_{l_1,l_2,|m_1|}^{Z_i,Z_j}(r_{ij}) \delta_{m_1,m_2} \tag{6}$$

Here, the dependency changes to $Z_i$, $Z_j$, and $r_{ij}$, indicating the two-center nature of the overlap integrals. Parameterizing these distance-dependent radial functions involves encoding atomic species with radial information and a simple MLP to learn the mapping to target scalars.

$$f_{l_1,l_2,|m|}(Z_i, Z_j, r_{ij}) = s^{Z_i,Z_j}_{l_1,l_2,|m_1|}(r_{ij})\delta_{m_1,m_2} \tag{7}$$

After getting these parameters, we use them to construct the overlap matrix aligned with the bond axis, and then rotate it back to the original orientation.

### 3.3 THE SLEM MODEL

The SLEM model architecture is illustrated in Fig. 2. The model maintains a set of features, including hidden features $\mathbf{x}^{ij,L}$, $\mathbf{V}^{ij,L}_{c,l}$, node features $\mathbf{n}^{i,L}_{c,l}$ and edge features $\mathbf{e}^{ij,L}_{c,l}$ of layer ($L$). Specifically, the hidden features consist of a scalar channel $\mathbf{x}^{ij,L}$ and a tensor channel $\mathbf{V}^{ij,L}_{c,l}$. The scalar channel is initialized as an embedded vector containing two-body information, including the atomic species and the radial distance of the atom pair. These initialized two-body radial features are then mapped by an MLP to the invariant SK parameters for overlap. For equivariant quantum operators, such as the Hamiltonian and density matrix, the scalar and tensor channels interact to generate an ordered atom-pair representations, which are used to construct local node and edge representations. After iterative updates, the representation is scaled by the statistical norm and biases, thereby achieving the final prediction.

#### 3.3.1 FEATURE INITIALIZATION

Firstly, the initial scalar hidden feature is computed from the two-body embeddings of atomic species $Z_i$ and $Z_j$, and the radial distance $r_{ij}$, as follows:

$$\mathbf{x}^{ij,L=0} = \mathbf{MLP}_{\text{2-body}}\left(\mathbf{1}(Z_i)||\mathbf{1}(Z_j)||\mathbf{B}(r_{ij})\right) \cdot u(r_{ij}) \tag{8}$$

Here, $||$ is vector concatenation. Atom species are embedded as one-hot vectors $\mathbf{1}(Z)$, and a set of trainable Bessel bases $\mathbf{B}(r_{ij})$ is utilized to encode the distance $r_{ij}$ between atoms $i$ and $j$. $u(r_{ij})$ are envelope functions (Batzner et al. (2022)) to add explicit radial dependency. Subsequently, the edge and hidden features are initialized as weighted spherical harmonics of relative edge vectors:

$$\begin{aligned}
\mathbf{V}^{ij,L=0}_{c,l} &= w_{c,l}\left(L_N\left(\mathbf{x}^{ij,L=0}\right)\right)\mathbf{Y}^{ij}_l \\
\mathbf{e}^{ij,L=0}_{c,l} &= w_{c,l}\left(\mathbf{x}^{ij,L=0}\right)\mathbf{Y}^{ij}_l
\end{aligned} \tag{9}$$

Here, the weights are learned from the initialized scalar hidden features $\mathbf{x}^{ij,L=0}$. Layer normalization $L_N$ ensures that the hidden tensor features have a balanced amplitude of each edge. The initial node features are then computed as linear transformations of the aggregated edge features:

$$\mathbf{n}^{i,L=0}_{c,l} = \mathbf{Linear}\left(\frac{1}{\sqrt{N_{\text{avg}}}}\sum_{j\in\mathcal{N}(i)}\mathbf{e}^{ij,L=0}_{c,l}\right) \tag{10}$$

Here $\mathcal{N}(i)$ and $N_{\text{avg}}$ are the neighbouring atoms and the average number of neighbours of atom $i$.

#### 3.3.2 SPEED UP TENSOR PRODUCT

To integrate the information from the equivariant features, the tensor product is employed in all updating blocks of the SLEM model. Generally, the tensor product in SLEM is performed with the concatenated equivariant features $\tilde{\mathbf{f}}^{ij}_{c,l}$ and the weighted projection of the edge shift vector $\boldsymbol{r}_{ij} = \boldsymbol{r}_i - \boldsymbol{r}_j$ on the spherical harmonics function $\mathbf{Y}^{ij}_l$. Formally:

$$\mathbf{f}^{ij}_{c_3,l_3} = \tilde{\mathbf{f}}^{ij}_{c_1,l_1} \otimes w^{ij}_{c_2,l_2}\mathbf{Y}^{ij}_{l_2} = \sum_{c_1,l_1,l_2}\tilde{w}^{ij}_{c_1,l_1,l_2}\sum_{m_1,m_2}C^{(l_3,m_3)}_{(l_1,m_1)(l_2,m_2)}f^{ij}_{c_1,l_1,m_1}Y^{ij}_{l_2,m_2} \tag{11}$$

Here, $\tilde{w}^{ij}_{c_1,l_1,l_2} = \sum_{c_2}w_{c_1,c_2,l_1,l_2}w^{ij}_{c_2,l_2}$ are edge-specific parameters for each tensor product operation. Performing such tensor products on high-order features is computationally intensive. Therefore, we applied the recently developed SO(2) (Passaro & Zitnick (2023)) convolution to reduce the computation and storage complexity from $O(l^6_{\text{max}})$ to $O(l^3_{\text{max}})$ which we refer to the Appendix 18.

### 3.3.3 HIDDEN UPDATES

To construct many-body interactions, as in Fig. 2(b), the node features $\mathbf{n}^i_{c,l}$ and hidden tensor features $\mathbf{V}^{ij}_{c,l}$ would be concatenated and doing tensor product with the projection coefficients of edge shift vector $\boldsymbol{r}_{ij}$ on the spherical harmonics functions. The operation is written formally as:

$$\tilde{\mathbf{V}}^{ij,L}_{c_3,l_3} = \left(\mathbf{n}^{i,L-1}||\mathbf{V}^{ij,L-1}\right)_{c_1,l_1} \otimes w^{ij,L}_{c_2,l_2} \mathbf{Y}^{ij}_{l_2} \tag{12}$$

Unlike most MPNN, the hidden states $\mathbf{x}^{ij}$ and $\mathbf{V}^{ij}_{c,l}$ in SLEM depend only on the local environment of centre atom $i$, the shift vector $\boldsymbol{r}_{ij}$, and the atomic type and coordinate informations of atom $j$. Such a design excludes neighbours of $j$ into hidden states.

After the tensor production, the output features $\tilde{\mathbf{V}}^{ij,L}_{c,l}$ will be passed through the gated non-linearity (Batzner et al. (2022)), and transformed by an "E3linear" (Geiger & Smidt (2022)) layer to mix up the information across different channels. The new hidden feature will be multiplied by the weights learned from normalized scalar features to explicitly include the radial information and return as the updated feature $\mathbf{V}^{ij,L}_{c,l}$. The scalar hidden features are updated by mixing the 0th order information from $\mathbf{V}^{ij,L}_{c,l}$ with a latent MLP, which is:

$$\mathbf{x}^{ij,L} = \mathbf{MLP}\left(\mathbf{x}^{ij,L-1}||\mathbf{V}^{ij,L}_{c,l=0}\right) \cdot u(r_{ij}) \tag{13}$$

The scalar hidden states $\mathbf{x}^{ij,L}$ incorporate an explicit decaying envelope function $u(r_{ij})$ and many-body interactions of scalar and tensor features. This formulation effectively captures the decay behaviour of each edge irrep feature as a function of radial distance.

### 3.3.4 NODE UPDATES

The strictly local node representation $\mathbf{n}^{ij,L}_{c,l}$ can be constructed naturally from the many-body interactive tensor features $\mathbf{V}^{ij,L}_{c,l}$. We follow the MPNN style to create the message from node $j$ to node $i$. Formally:

$$\mathbf{m}^{ij,L}_{c_3,l_3} = \left(\mathbf{n}^{i,L-1}||\mathbf{V}^{ij,L}\right)_{c_1,l_1} \otimes w^{ij,L}_{c_2,l_2} \mathbf{Y}^{ij}_{l_2} \tag{14}$$

Here again, we exclude the neighbouring information of atom $j$ in $\mathbf{m}^{ij,L}_{c_3,l_3}$ via the partial updates of $\mathbf{V}^{ij,L}_{c,l}$, while maintaining necessary interactions. Each message then is passed through a gated activation and E3Linear layer, weighted separately by weights learnt from the hidden scalar features, and aggregated to update the node feature by:

$$\mathbf{n}^{ij,L}_{c_3,l_3} = \alpha \cdot \mathbf{n}^{ij,L-1}_{c_3,l_3} + \frac{\sqrt{1-\alpha^2}}{N_{avg}} \sum_{j \in \mathcal{N}(i)} w^{ij}_{c_3,l_3} \mathbf{m}^{ij,L}_{c_3,l_3} \tag{15}$$

Here $\alpha$ ranged from 0 to 1. The weights here differ from those in hidden updates as they are directly learnt from $\mathbf{x}^{ij,L}$ without normalization. Therefore, the absolute radial decay is enforced in the weights, providing a strong prior that the messages from atoms at shorter distances are generally more significant. Meanwhile, the update of $\mathbf{x}^{ij,L}$, and consequently $w^{ij}_{c_3,l_3}$, depends on the features $\mathbf{V}^{ij,L}$ and $\mathbf{n}^{i,L-1}$, as shown in Eq.13. This structure aligns with the equivariant graph attention mechanism (Liao & Smidt; Liao et al.) which has demonstrated powerful expressibility in various tasks. Here $w^{ij}_{c_3,l_3}$ corresponding to an attention score computed from $\mathbf{V}^{ij,L}$ and $\mathbf{n}^{i,L-1}$. Therefore, through this update, the dependencies of node features are strictly local.

### 3.3.5 EDGE UPDATES

The locality of edge features $\mathbf{e}^{ij,L}_{c,l}$ can be naturally preserved as long as the node features are strictly local. By mixing the information of node features on both sides, localized edge updates can be formulated as follows:

$$\tilde{\mathbf{e}}^{ij,L}_{c_3,l_3} = \left(\mathbf{n}^{i,L-1}||\mathbf{V}^{ij,L}||\mathbf{n}^{j,L-1}\right)_{c_1,l_1} \otimes w^{ij,L}_{c_2,l_2} \mathbf{Y}^{ij}_{l_2} \tag{16}$$

Table 1: MAE (in meV) for Hamiltonian matrix predictions using SLEM and other methods on materials with LCAO basis up to $d$ orbitals. Numbers in parentheses indicate parameter count.

| Material | SLEM | | DeepH-E3 | | HamGNN |
|---|---|---|---|---|---|
| | (0.7M) | (4.5M) | ( 1.0 M) | ( 4.5M) | (2.8M) |
| $MoS_2$ | **0.34** | **0.14** | 0.46 | 0.55 | 1.20 |
| Graphene | **0.26** | **0.14** | 0.40 | 0.28 | 0.35 |
| Si(300K) | **0.10** | **0.07** | 0.16 | 0.10 | 0.19 |

Systems with LCAO-basis up to $d$-orbitals

Table 2: MAE (in meV) for Hamiltonian matrix predictions using SLEM and DeepH-E3 models (Gong et al. (2023)) on materials with LCAO basis up to $f$ and $g$ orbitals. Numbers in parentheses indicate parameter count. The MAE of $HfO_2$ in DeepH-E3 is absent due to out-of-memory errors.

| Material | SLEM | DeepH-E3 |
|---|---|---|
| | (1.7M) | (1.9M) |
| GaN | **0.21** | 0.87 |
| $HfO_2$ | **0.28** | - |

Systems with LCAO-basis up to $f$ and $g$-orbitals

Similarly, the updated edge features are processed via a gated activation and an E3Linear layer, and then multiplied with weights learnt from the hidden scalar features (without normalization) as:

$$\mathbf{e}_{c_3,l_3}^{ij,L} = \alpha \cdot \mathbf{e}_{c_3,l_3}^{ij,L-1} + \sqrt{1-\alpha^2} \cdot w_{c_3,l_3}^{ij} \tilde{\mathbf{e}}_{c_3,l_3}^{ij,L} \tag{17}$$

The following section focuses on validating the effectiveness of this framework via learning equivariant DFT Hamiltonians, density matrices and overlap.

## 4 RESULTS

### 4.1 BENCHMARK THE ACCURACY AND DATA-EFFICIENCY

Using diverse datasets includes up to $g$ orbitals, we evaluate our model's performance in fitting Hamiltonian, density matrix, and overlap matrix. For Hamiltonian, we use systems including 2D systems of monolayer $MoS_2$ and graphene from existing datasets (Li et al. (2022)), as well as 3D bulk silicon generated for this study. To test SLEM's capability with high-order tensors, we also train the models on generated datasets of bulk GaN and $HfO_2$ systems, which include $f$ and $g$ orbitals. Structures in the Si, GaN, and $HfO_2$ systems are sampled via molecular dynamics using neural network potentials (Wang et al. (2018)). For density matrix and overlap matrix evaluations, we focus exclusively on the datasets of Si, GaN, and $HfO_2$ datasets, as the reported datasets for $MoS_2$ and graphene lack density and overlap matrix data. Our model supports an atom-specific $r_{cut}$ setting, each value corresponding to the radial basis cutoff of the numerical atomic orbitals, which aligns with the setting in the compared method.

Table 1 presents a comparison of mean absolute error (MAE) values in Hamiltonian prediction for graphene, $MoS_2$, and Si systems, whose LCAO basis extends up to $d$ orbitals, among the SLEM, DeepH-E3 (Gong et al. (2023)), and HamGNN (Zhong et al. (2023)) methods. Our SLEM model achieves state-of-the-art accuracy, exhibiting the lowest MAE across all systems. Notably, this high accuracy is achieved with a relatively small model size of only 0.7 million (M) trainable parameters. Furthermore, we extended our comparison between the SLEM and DeepH-E3 methods to include GaN and $HfO_2$ systems, where the LCAO basis extends up to $f$ and $g$ orbitals, respectively. As

Table 3: MAE for predicting density matrix using SLEM on materials with LCAO basis up to $d$, $f$, and $g$ orbitals. Model settings align with those in Table 1.

| SLEM density matrix model | | | |
|---|---|---|---|
| Materials | Silicon | GaN | $HfO_2$ |
| MAE | 8.9e-5 | 2.3e-5 | 3.9e-5 |

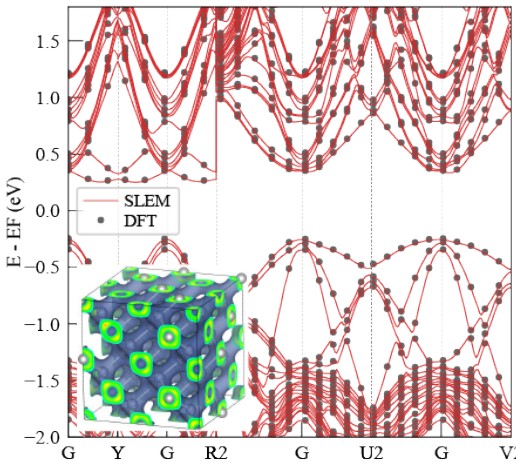

Figure 3: Comparison of band structures for a Si MD trajectory snapshot: SLEM prediction vs. DFT calculation. Predicted band structures are obtained from either diagonalization of the predicted Hamiltonian or NSCF DFT calculation using predicted charge density, yielding indistinguishable results. Inset: Visualization of charge density distribution for the same structure.

shown in Table 2, the SLEM model consistently presents the lowest MAE, further demonstrating its high accuracy and versatility across different orbital complexities. Fig. 3 illustrates the band structure of silicon structures computed directly from the SLEM-predicted Hamiltonian, where the eigenvalues are indistinguishable from those obtained using DFT. For the density matrix, fitting results are presented in Table 3. The results demonstrate very high accuracy (order of 1e-5), approaching the machine precision limit of float32 numbers. In Fig. 3, we use the trained model to predict the density matrix and visualize its real-space distribution. This capability is particularly important for applications such as charge distribution analysis or tracking electron transfer. Furthermore, the predicted density can be directly used for non-self-consistent field (NSCF) DFT calculations. The resulting band structure for silicon, as an example, is highly accurate and matches the DFT output, with a MAE of only 1.09 meV in eigenvalues compared to self-consistent DFT results. The overlap matrix, represented by invariant SK parameters in our SLEM model, achieves exceptionally high accuracy as demonstrated in Table 4, approaching the machine precision limit for float32 numbers. Notably, our simplified parameterization enables this high accuracy with only a minimal increase in model complexity. For instance, in a silicon model designed to fit only the Hamiltonian, our typical parameter count is 0.7 M. The inclusion of overlap matrix prediction adds merely 0.01 M parameters, which is about 1.4% in total model size.

Additionally, the strict localization scheme of our SLEM model confers superior data efficiency, requiring fewer DFT-calculated data points for training. To quantify this efficiency, we conducted an experiment using randomly split subsets of the original data, comprising 20%, 40%, 60%, and 80% of the full training set. We trained both the SLEM model and DeepH-E3 method on these subsets and evaluated their performance on consistent validation sets. Results in Table 5 demonstrate SLEM's high accuracy across all subsets, thus highlighting its remarkable data efficiency. This data efficiency implies that SLEM users can generate smaller, more cost-effective training sets. Moreover, SLEM's excellent data efficiency and transferability make it particularly well-suited as a backbone for developing universal DFT models, especially for systems involving heavy elements.

Table 4: MAE for predicting overlap matrix using SLEM's parameterization on materials with LCAO basis up to $d$, $f$, and $g$ orbitals. Model settings align with those in Table 1.

| | SLEM overlap prediction | | |
|---|---|---|---|
| Materials | Silicon | GaN | $HfO_2$ |
| MAE | 5.6e-5 | 4.7e-5 | 6.3e-5 |

Table 5: Comparison of validation MAE (in meV) for SLEM model and DeepH-E3 (Gong et al. (2023)) method trained on randomly split datasets (Li et al. (2022)) with varying training ratios. Model settings align with those in Table 1.

| | MoS2 | | | | |
|---|---|---|---|---|---|
| Partition | 100% | 80% | 60% | 40% | 20% |
| SLEM | **0.34** | **0.37** | **0.39** | **0.37** | **0.37** |
| DeePH-E3 | 0.46 | 0.72 | 0.84 | 1.03 | 1.46 |
| | Graphene | | | | |
| Partition | 100% | 80% | 60% | 40% | 20% |
| SLEM | **0.26** | **0.26** | **0.27** | **0.21** | **0.26** |
| DeePH-E3 | 0.40 | 0.30 | 0.33 | 0.36 | 0.60 |

## 4.2 EFFICIENCY AND SCALABILITY

In materials science, chemistry, and biology, many significant properties emerge in systems containing heavy atoms. These heavy atoms introduce high-order spherical tensors when representing quantum operators. Scaling to such systems is challenging due to the computational complexity of tensor products used to construct complex spherical tensors, which scales as $O(l^6)$. Conventional tensor production methods struggle with training and inference on systems containing heavy atoms, making it difficult to model these important phenomena efficiently. Moreover, inferring large material systems while training with small structures is particularly valuable, which requires parallelising the model inference by assigning partitions of the large atomic structure to multiple GPU workers. However, most current models struggle with this task. As the receptive fields expand through iterative graph updates, the minimum size of each partitioned subgraph increases, reducing the effectiveness of such partitioning. The SLEM model addresses these challenges by efficiently constructing high-order tensor products and assisting parallelization through its strict locality.

For efficiency, the implementation of SO(2) convolution reduces the tensor product computational complexity from $O(l^6)$ to $O(l^3)$, which is then further reduced by the parallelization of matrix operations to nearly $O(l)$ benefiting from PyTorch. Figure 4 compares the wall time and GPU memory consumed by tensor product operations using SO(2) convolution versus the conventional method employed in DeepH-E3 (Gong et al. (2023)) and E3NN (Geiger & Smidt (2022)). The SO(2) convolution approach demonstrates remarkable superior efficiency, enabling our method to handle all possible basis choices in LCAO DFT. We also evaluated memory usage and training time for typical systems, comparing our model with DeepH-E3. The results, displayed in Fig. 5, show that our model consistently outperforms in both metrics. Notably, the advantage becomes more pronounced as the basis set size increases, highlighting our method's scalability. Additionally, the SLEM model's strict locality design significantly enhances parallelization. This localized approach allows for dividing the atomic graph into sub-graphs, enabling independent computation of node and edge features on separate devices. This is extremely important when expanding DFT simulation to large systems. In practice, for $HfO_2$ with $4s2p2d2f1g$ basis, a typical model of 1.7M parameters can predict the quantum operators for up to $10^3$ atoms on devices with 32GB memory. Despite linear scaling, the inference on a system with $10^4 \sim 10^7$ would require over 300 GB of memory, necessitating parallelization across multiple GPUs. Therefore, a strictly localized model such as SLEM holds significant potential for expanding simulation system sizes.

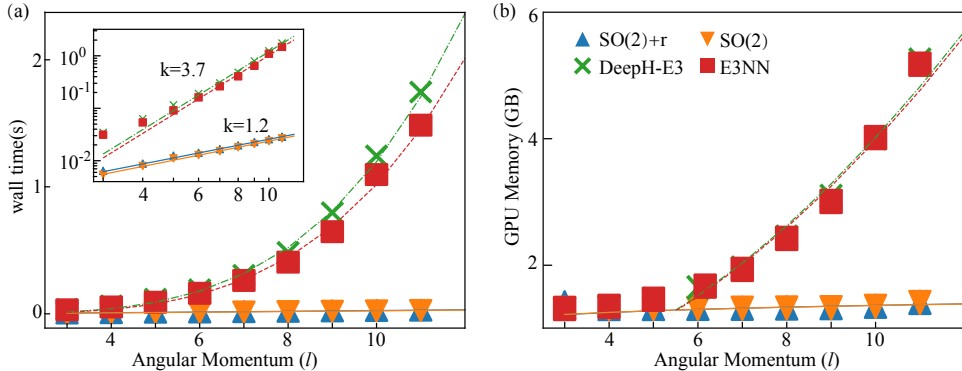

Figure 4: Comparison of time and memory consumption for different tensor-product implementations. (a) Time consumption vs. angular momentum ($l$) for different models, including the SO(2)-based SLEM model (triangles) with and without radial part ($r$), DeepH-E3 (cross) (Gong et al. (2023)), and E3NN (square) (Geiger & Smidt (2022)) models. Inset: Log-scale fit with slopes of 1.2 for the SLEM model and 3.7 for the other two models. (b) Memory consumption vs. $l$. The SLEM model demonstrates over two orders of magnitude improvement in both time and memory efficiency compared to DeepH-E3 and E3NN.

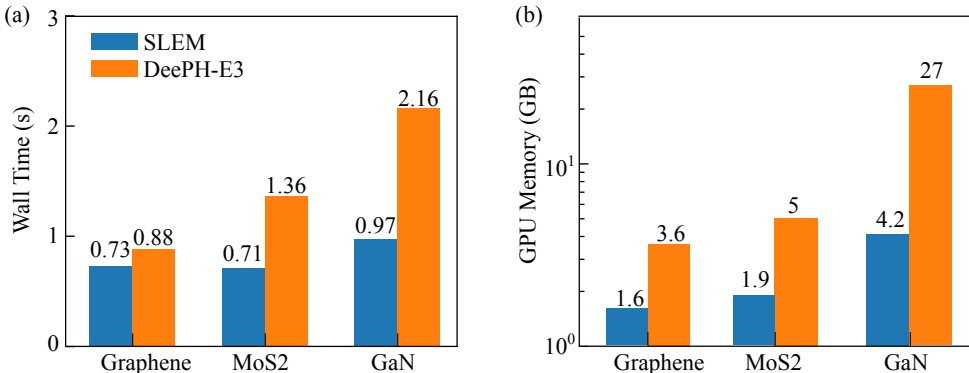

Figure 5: Comparison of training time per iteration and memory consumption with SLEM and DeepH-E3 (Gong et al. (2023)) models.

## 5 CONCLUSION

This work presents SLEM (Strictly Local Equivariant Message-passing) model, a novel approach for predicting quantum operator representations in materials science. By employing a strict locality design and integrating SO(2) convolution, SLEM achieves state-of-the-art performance in predicting Hamiltonians, density matrices, and overlap matrices for diverse materials, including systems with heavy atoms. The model's efficiency and scalability open new possibilities for large-scale quantum simulations and high-throughput materials discovery. Notably, SLEM's locality enables efficient parallelization through atomic graph partitioning, potentially extending its applicability to extremely large systems at device-level scales. SLEM's intrinsic support for multiple quantum operators, coupled with its novel overlap matrix parameterization, significantly reduces computational costs and dependence on post-training DFT software. The model demonstrates superior data efficiency and transferability, making it particularly well-suited for developing universal DFT models, especially for systems involving heavy elements. These advancements position SLEM as a powerful tool for simulating complex systems in materials science and computational chemistry. Future work will focus on developing robust sampling methodologies for active learning and integrating SLEM with existing software ecosystems to fully leverage its capabilities in real-world applications, further advancing the field of computational materials science.

CODE AVAILABILITY

The implementation of the SLEM model and its semi-local variant (named LEM) are open-source and accessible via the DeePTB GitHub repository. To facilitate the integration of DFT outputs with machine learning models, we have developed and released a supplementary tool, dftio. This tool enables parallel parsing of DFT outputs into a machine-readable data format, enhancing the efficiency of data preparation for model training and analysis.

ACKNOWLEDGMENTS

We appreciate Mingkang Liu for his contribution in validating the SLEM/LEM model on QH9 datasets during the review process. The results are presented in the Appendix A. We sincerely acknowledge his efforts and regret that we were unable to include him as a co-author due to ICLR authorship policies. We also appreciate insightful discussions with Siyuan Liu, Feitong Song, Jijie Zou, and Duo Zhang. We acknowledges the funding support from AI for Science Institute, Beijing (AISI). The computational resource is provided by Bohrium Cloud Platform from DP technology.

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

## A BENCHMARK ON QH9

We also train our model on the open molecule Hamiltonian and Overlap dataset, QH9 (Yu et al. (2024)), to test our method's accuracy and transferability on a large dataset. We compare our result with the reported benchmark QHNet, which shows considerable accuracy improvement, as displayed in Table. 6. We should notice that our model performs well in the Hamiltonian learning tasks, by decreasing 1/3 of the error reported previously. The density matrix is inferred from the learned Hamiltonian since the data is not available in QH9. Our accuracy is only 1/3 of the compared one.

QH9 is an extended electronic structure dataset based on QM9 (Ruddigkeit et al. (2012); Ramakrishnan et al. (2014)), a well-known benchmark for molecular property prediction. QH9 contains 13,081 static structures and 2998 molecular trajectories. We test our method based on the static out-of-distribution(OOD) tasks defined by QHNet's paper. In this task, the testing set has a very different molecular structure and atom number from the ones in the training set. The dataset is split as 104,001/17,495/9,335. As discussed in Appendix G, we used a semi-local variant of SLEM to prevent the possible break of strict locality in the molecular system.

Table 6: MAE for Hamiltonian matrix (H) and density matrix (D) prediction.

| Method | $\mathbf{H}[10^{-6}E_h]$ | | | **D** |
|---|---|---|---|---|
| | all | non-diag | diag | all |
| LEM | **56.57** | **44.86** | 155.24 | **0.0219** |
| QHNet | 83.22 | 80.26 | 135.63 | 0.0643 |

## B ON THE STRICT LOCALITY APPROXIMATION

Here we want to further discuss the strict locality approximation made in our method. In the quantum operator learning task, we express the physical operators using an LCAO basis. In this expression, the theoretical framework of DFT fundamentally supports the strict locality approximation. In LCAO-based DFT calculations, the Hamiltonian matrix elements exhibit inherent locality. This locality manifests through two mechanisms: the spatial decay of LCAO basis functions, the electron neuralized conditions of the physics system, and electronic screening effects especially in condensed matter systems. These effects make the dependency of the quantum operator elements approximately local. For periodic systems, particularly metals and semiconductors, electronic screening effectively reduces long-range Coulomb interactions. The screening length varies by material type around 4 Å in semiconductors like silicon, and Germanium, and smaller in insulators such as 2.76 Å in Diamond Therefore, we are safe to choose a cutoff radius to match the sum of basis orbital radii, adequately captures the correct physical relations. Taking the Hamiltonian matrix elements as an example:

$$V_{ij}^H = \int \phi_i^*(r - R_A)\left[V_H(r) + V_{ext}(r) + V_{xc}(r)\right]\phi_j(r - R_B)dr$$

The locality is a combination of the LCAO basis, the Hartree term, the external potential, and xc functional (we ignore the kinetical term for its local nature). Therefore, we need to discuss the localization as a collective effect, which is contributed by the following properties:

**The decaying behaviour of the LCAO basis.** For all LCAO bases (whether GTO, NAO, or Slater-like), the radial part all decays rapidly with distance. Therefore, the long-range term in $V_H, V_{ext}, V_{xc}$'s contribution in the integration (that decays as 1/r) would be decreased fastly (often exponentially such as in GTO or Slater-like orbital).

**The system's electronic neutralization condition.** The electronic neutralization condition requires that the long-term part of the Hartree term and the external term cancel each other. To derive this, we assign the electron density to the atomic center, by writing as:

$$n(r) = \sum_I n_I(r - R_I)$$

Then the sum of $V_H(r), V_{ext}(r)$ would be:

$$-\sum_I \frac{Z_I}{|r - R_I|} + \sum_I \int \frac{n_I(r' - R_I)}{|r - r'|} dr'$$

When $|r - R_I|$ is large, we can approximate $|r - r'| \approx |r - R_I|$, therefore:

$$-\frac{Z_I}{r - R_I} + \int \frac{n_I(r' - R_I)}{|r - r'|} dr' \approx -\frac{Z_I}{|r - R_I|} + \frac{Z_I}{|r - R_I|} = 0$$

**The screening effect.** we refer to Huckel & Debye (1923); Resta (1977); Ninno et al. (2006) the study of screening radius $R_s$, which describes the system's electrostatic potential's reaction to the vibration of charges.

In Resta (1977), the screening radius of typical insulator and semiconductor systems is reported as: Diamond: 2.76 a.u, Silicon: 4.28 a.u. and Germanium: 4.71 a.u. In Ninno et al. (2006), for some nanoparticles, the $R_s$ is reported as: Si191H148: 5.36 a.u. and Ge191H148: 5.61 a.u.

These effects altogether motivate us to revise the current method for machine learning electronic structures and design SLEM, as a strict localized model suitable for learning the correct mapping from structures to quantum operators with the correct physical priors—a more reliable and data-efficient method. Meanwhile, we also notice the existing limitation of this method in some special systems, we refer to the discussion in Appendix G for detail.

## C   COMPARISON OF SLEM WITH MPNN

**Message-passing Neural Networks**   In the message-passing scheme, atoms are treated as nodes in graphs, with bonds to neighbouring atoms represented as connected edges within a specified cutoff radius. The embedded atomic features are processed by trainable functions, generating messages from each edge to update the embeddings of central atoms. Formally, the MPNN framework can be summarized as follows:

$$\mathbf{e}^{ij,L} = \mathcal{N}_L\left(\mathbf{n}^{i,L-1}, \mathbf{n}^{j,L-1}, \mathbf{e}^{ij,L-1}\right)$$
$$\mathbf{m}^{ij,L} = M_L\left(\mathbf{n}^{i,L-1}, \mathbf{n}^{j,L-1}, \mathbf{e}^{ij,L}\right)$$
$$\mathbf{n}^{i,L} = U_L\left(\mathbf{n}^{i,L-1}, \sum_{j \in \mathcal{N}(i)} \mathbf{m}^{ij,L}\right)$$

Here, $\mathbf{e}^{ij,L}$ represents the edge features, $\mathbf{m}^{ij,L}$ denotes the messages, and $\mathbf{n}^{i,L}$ indicates the node features at layer $L$. $\mathcal{N}_L$, $M_L$, and $U_L$ are the trainable functions for the edge, message, and node updates. $\mathcal{N}(i) = \{j | r_{ij} < r_{cut}\}$ indicates all the neighbour atoms for atom $i$, with $r_{cut}$ being the predefined cutoff. This updating framework allows for the construction of many-body interactions and long-term dependencies, leading to strong performance across various applications. Here we

reframe the updating rules of SLEM with the MPNN framework, which looks like this:

$$\mathbf{V}^{ij,L} = \mathcal{V}_L \left( \mathbf{n}^{i,L-1}, \mathbf{V}^{ij,L-1} \right)$$

$$\mathbf{e}^{ij,L} = \mathcal{N}_L \left( \mathbf{n}^{i,L-1}, \mathbf{V}^{ij,L}, \mathbf{n}^{j,L-1}, \mathbf{e}^{ij,L-1} \right)$$

$$\mathbf{m}^{ij,L} = M_L \left( \mathbf{n}^{i,L-1}, \mathbf{V}^{ij,L} \right)$$

$$\mathbf{n}^{i,L} = U_L \left( \mathbf{n}^{i,L-1}, \sum_{j \in \mathcal{N}(i)} \mathbf{m}^{ij,L} \right)$$

Here $\mathcal{V}_L$ is the neural network for hidden feature construction. This update scheme constructs many-body interactions to build equivariant edge and node features while preserving the absolute locality by excluding atoms outside the constant cutoff radius. Such a scheme, in principle, would have much better transferability, data efficiency, and scalability when we have a strong prior that the input and output are dependent locally.

## D   MORE ON TENSOR PRODUCT

To integrate the information from the equivariant features, the tensor product is employed in all updating blocks of the SLEM model. Generally, the tensor product in SLEM is performed with the concatenated equivariant features $\tilde{\mathbf{f}}^{ij}_{c,l}$ and the weighted projection of the edge shift vector $\boldsymbol{r}_{ij} = \boldsymbol{r}_i - \boldsymbol{r}_j$ on the spherical harmonics function $\mathbf{Y}^{ij}_l$. Formally:

$$\mathbf{f}^{ij}_{c_3,l_3} = \tilde{\mathbf{f}}^{ij}_{c_1,l_1} \otimes w^{ij}_{c_2,l_2} \mathbf{Y}^{ij}_{l_2} = \sum_{c_1,l_1,l_2} \tilde{w}^{ij}_{c_1,l_1,l_2} \sum_{m_1,m_2} C^{(l_3,m_3)}_{(l_1,m_1)(l_2,m_2)} f^{ij}_{c_1,l_1,m_1} Y^{ij}_{l_2,m_2} \quad (18)$$

Here, $\tilde{w}^{ij}_{c_1,l_1,l_2} = \sum_{c_2} w_{c_1,c_2,l_1,l_2} w^{ij}_{c_2,l_2}$ are edge-specific parameters for each tensor product operation. Performing such tensor products on high-order features is computationally intensive. Therefore, we applied the recently developed SO(2) (Passaro & Zitnick (2023)) convolution to simplify, reducing the computation and storage complexity from $O(l^6_{\max})$ to $O(l^3_{\max})$. The simplification idea is intuitive. $\mathbf{Y}^{ij}_{l_2,m_2}$ are sparse tensors if rotated to align with the edge $ij$, which is nonzero only for $m_2 = 0$. Therefore, it is easier to compute the tensor production in the direction of edge $ij$, and rotate inversely the output afterwards. This step removes the $m_2$ index from the summation in Eq. 18. Furthermore, considering the Clebsch-Gordan coefficients with $m_2 = 0$, we find that $C^{(l_3,m_3)}_{(l_1,m_1)(l_2,0)} = 0$ except for $m_3 = \pm m_1$. This allows further reduction of the summation in Eq. 18 by replacing $\pm m_1$ with a single index $m$. Then the operations can be reformulated formally as:

$$\begin{pmatrix} f^{ij}_{c,l,m} \\ f^{ij}_{c,l,-m} \end{pmatrix} = \sum_{c',l'} \begin{pmatrix} w_{c,c',l',m} - w_{c,c',l',-m} \\ w_{c,c',l',-m} \quad w_{c,c',l',m} \end{pmatrix} \cdot \begin{pmatrix} \tilde{f}^{ij}_{c',l',m} \\ \tilde{f}^{ij}_{c',l',-m} \end{pmatrix}$$

This represents a linear operation on $\tilde{f}^{ij}_{c',l',m'}$. By employing this method, high-order tensor products for $l = 8, 9$ and even 10 can be efficiently calculated, which is essential for heavy element systems where the $f$, $g$ or even higher orbitals are used in the LCAO basis for DFT calculations. Finally, the weights for the new SO(2) tensor product method are multiplied by edge-specific parameters, where $w^{ij}_{c',l'}$ are mapped by an MLP from hidden scalar features $\mathbf{x}^{ij,L}$ as $\tilde{w}^{ij}_{c,c',l',m} = w_{c,c',l',m} w^{ij}_{c',l'}$. This powerful and efficient tensor product layer facilitates the construction of local interactive updates of the features.

## E   DATA GENERATION

In this section, we discuss the data generation process used in experiments of this work. The data sampling process includes the Hamiltonian, overlap and density matrix of materials Si, GaN, HfO$_2$.

First, we perform an *ab initio* molecular dynamic simulation based on a neural network force field using DeePMD (Wang et al. (2018)) and Lammps (Thompson et al. (2022)). A typical input file of the lammps sampling looks like:

```
variable        NSTEPS          equal 500000
variable        THERMO_FREQ     equal 100
variable        DUMP_FREQ       equal 1000
variable        TEMP            equal 300
variable        PRES            equal 1.00
variable        TAU_T           equal 0.10
variable        TAU_P           equal 0.50

units           metal
boundary        p p p
atom_style      atomic

neighbor        1.0 bin

read_data       conf.lammps-data
mass            1 28.085

pair_style      deepmd graph.pb
pair_coeff      * * Si

thermo_style    custom step temp pe ke etotal press vol lx ly lz xy xz yz
thermo          ${THERMO_FREQ}
restart         1000000 dpgen.restart

velocity        all create ${TEMP} 449414
fix             1 all nvt temp ${TEMP} ${TEMP} 0.1
dump            1 all custom ${DUMP_FREQ}  eq.lammpstrj id type x y z vx vy vz

timestep        0.001

run     100000

undump 1
dump            2 all custom ${DUMP_FREQ}  sample.lammpstrj id type x y z vx vy vz
run             ${NSTEPS} upto
```

The samples are taken after $10^5$ steps when the system reaches equilibrium.

After the configuration sampling, we then applied ABACUS (Li et al. (2016)) to compute the quantum operators, including the Hamiltonian, density and overlap matrix of each configuration. Among them, 150 frames are randomly split as the training set, and 30 frames are used for testing. We use SG15 ONCV numerical atomic orbitals, pseudopotentials (Hamann (2013)), and PBE functionals (Ernzerhof & Scuseria (1999)) for the DFT calculation. A typical DFT calculation input looks like:

```
INPUT_PARAMETERS
# Created by Atomic Simulation Enviroment
ntype                                   1
ecutwfc                                 100
scf_nmax                                100
smearing_method                         gaussian
smearing_sigma                          0.002
basis_type                              lcao
ks_solver                               genelpa
mixing_type                             pulay
mixing_beta                             0.7
scf_thr                                 1e-07
out_chg                                 1
symmetry                                1
calculation                             scf
out_band                                1
```

```
force_thr                               0.001
out_stru                                1
kspacing                                0.08
lspinorb                                0
out_wfc_lcao                            0
dft_functional                          pbe
out_mat_hs2                             True
```

After the calculation converged for all, we processed the data format into machine-learning readable datasets and started training. The dataset used in this work is uploaded in the opensource platform AISquare via this link: `https://www.aissquare.com/datasets/detail?pageType=datasets&name=Quantum_Operator_Dataset&id=286`.

## F   CUTOFF SELECTION

The cutoff used in the SLEM model is important since it decides the size of the atom and bond environment to consider when learning the map from the atomic structure to the quantum operators. There are two cutoff concepts in the SLEM model. One is the cutoff for the bond $r_b$, which is used to build the one-to-one correspondence of the graph structure of atomic data to the diagonal and off-diagonal blocks of the quantum operators. Another one is the cutoff for environment $r_e$.

For $r_b$, since the bond whose radial distance beyond $r_b$ would all be considered 0, this value should be fixed according to the LCAO orbital used in DFT data generation. For numerical atomic orbitals (NAOs), the cutoff is defined clearly in orbital files or DFT software, therefore, we just double (since two orbitals constitute a bond) and assign the value to each atom as there $r_b$. For Gaussian-type orbitals (GTOs) and Slater-type orbitals (STOs), since they decay exponentially along radial distance, we often set some precision threshold during data processing and find the longest bond in off-diagonal blocks with a value larger than the preset threshold. The length of the bond will be considered as $r_b$.

The environmental cutoff $r_e$ is very system dependent. This decides how large atomic/bond environments relate to the mapping. This should follow the locality behaviour of the physical system. For example, in typical periodic crystals such as semiconductors (Si, GaN, etc.), medals (Graphene), and insulators ($HfO_2$) the $r_e$ are quite local, therefore, it is safe to set the $r_e = r_b$. However, when systems experience strong nonlocality behaviour, $r_e$ should be very large. One should carefully balance the efficiency of choosing a large $r_e$ that ensures strict locality and the usage of a semi-local or MPNN-based method.

## G   LIMITATION

The SLEM model, benefiting from the Strictly localized design, performs better in accuracy and transferability as demonstrated in various datasets. We acknowledge that such a localized hypothesis is most suitable for describing periodic systems, which is commonly adopted in physics, chemistry and material science research. For confined systems such as molecules, the absence of screening effect could lead to long-term dependency that is uncovered within prefixed cutoff. In these cases, the SLEM model wouldn't perform as well as it does in periodic ones. For generality, we also proposed a semilocal model called LEM (Localized Equivarient Message-Passing), where the interaction between distant atoms is included but decays exponentially with their distance, with a trainable decay factor. All the designed models are ready to use in our GitHub repository: `https://github.com/deepmodeling/DeePTB`.

## H   FUTURE INVESTIGATION

While SLEM demonstrates efficacy across various applications, several areas warrant further investigation:

- Sampling Methodology for Active Learning: Scientific computation tasks require high confidence in calculated results, which can be challenging for data-driven approaches. Developing a robust sampling workflow for active learning is essential for reliability. While techniques like uncertainty-driven sampling with Gaussian regression (Vandermause et al. (2020)) or model ensembles (Zhang et al. (2020)) have addressed confidence issues in machine learning force fields, how to design a sampling workflow specifically for quantum operator models remains an open question.

- Software Integration for Post-Processing: Integrating the model with existing software is vital, especially for high-throughput calculations and large atomic systems beyond conventional DFT capabilities. An efficient, parallelizable solver for extracting physical quantities from the model's predictions is highly beneficial. While some software based on stochastic techniques (Li et al. (2023); João et al. (2020)) shows promise, an open-sourced and highly-optimized solution for non-orthogonal bases remains unavailable.

Future research addressing these challenges will further enhance the applicability and reliability of SLEM and similar quantum operator models in computational materials science and chemistry. We are excited and look forward to seeing these happen.

