# OpenReview forum: "Learning local equivariant representations for quantum operators"
_ICLR.cc/2025/Conference — ICLR 2025 Spotlight_

### Official Review · Reviewer_nQRm · 2024-10-27

**Soundness:** 3
**Presentation:** 3
**Contribution:** 3
**Rating:** 8
**Confidence:** 3

**Summary:**

This paper introduces SLEM, a model for predicting quantum operator matrices that achieves state-of-the-art accuracy and improves computational efficiency. It is strictly local and uses a SO(2) convolution technique, enabling it to handle systems with heavy atoms and high angular momentums while maintaining efficiency. The work shows experimental validation across various materials, demonstrating better performance in both accuracy and computational cost compared to existing methods

**Strengths:**

The main innovation and strength of this paper is combining strict locality, with an architecture heavily inspired in Allegro, with the SO(2) convolution trick, offering a compelling solution to handling heavy atoms and large systems efficiently while maintaining accuracy. The work can be considered well validated through comprehensive benchmarks, and they demonstrate the previous claims of both improved accuracy and reduced computational costs compared to state-of-the-art methods. The authors also create new datasets, and making them available would improve this paper's impact

**Weaknesses:**

A major limitation of the paper is its insufficient demonstration of transferability. While the authors show good performance on individual systems, they train and evaluate on the same type of material (e.g., training on Si and testing on Si configurations). There's no evaluation of cross material transferability. For instance, training on light elements and testing on heavy elements, or training on one crystal structure and testing on another. Also, the parallelization benefits, while promising (and I believe them, since they should be inherited from Allegro), are mainly theoretical with regards to this paper. The paper lacks concrete scaling studies on large systems or benchmarks on multiple GPU setups that would validate their claims about improved parallelizability.

**Questions:**

1) Are there specific cases where you expect the strict locality assumption to break down? The method perhaps could not handle systems with strong electronic correlations.
2) It is not clear from the paper (at least I could not find it) which is the cutoff used, or if different cutoffs were used for different systems. This could enable comparing the receptive field of the model proposed in this paper and other approaches using message-passing.
3) Evaluating the model performance in a transferability task would enhance significantly the submission.
4) Evaluating the scaling of the model in a multigpu setting would also enhance significantly the submission.

---

> ### Author Response · Authors · 2024-11-22
>
> We appreciate the reviewer's kind comments about our work. We especially, thank the recognition for the experiment that is well validated. Your kind applause encourages us. We are also very happy to see the construction suggestions including the data openness, the transferability test and the multi-GPU inference. These suggestions help improve our manuscript and provide guidance for further research. We have uploaded all the data used in this work in the “AIS Square” with the link of  https://www.aissquare.com/datasets/detail?pageType=datasets&name=Quantum_Operator_Dataset&id=286 . We hope this can increase the reproducibility. In the following discussion, we will address the questions and suggestions listed one by one.
>
> ***Q1. Are there specific cases where you expect the strict locality assumption to break down? The method perhaps could not handle systems with strong electronic correlations.***
>
> The theoretical framework of DFT fundamentally supports the strict locality assumption in our method. In LCAO-based DFT calculations, the Hamiltonian matrix elements exhibit inherent locality. This locality manifests through two mechanisms: the spatial decay of LCAO basis functions and electronic screening effects in condensed matter systems. The only difference lies in how large the local spread is.
> For periodic systems, particularly metals and semiconductors, electronic screening effectively reduces long-range Coulomb interactions. The screening length varies by material type around 4 Å in semiconductors like silicon, and Germanium, and smaller in insulators such as 2.76 Å in Diamond Therefore, we are safe to choose a cutoff radius to match the sum of basis orbital radii, adequately captures these screening effects.
> Regarding strongly correlated systems, it's important to distinguish between method limitations and framework limitations. While these systems present challenges, the limiting factor is the accuracy of the underlying DFT calculation rather than our locality assumption. Our method can still effectively learn the DFT mapping, inheriting both its capabilities and limitations.
> For molecular systems where screening effects are weaker, we acknowledge the need for larger cutoff radii. This motivated our development of the LEM variant, which introduces trainable but exponentially decaying environmental dependencies to handle the weaker screening effects in molecular systems. Our benchmark results on the QH9 dataset demonstrate the effectiveness of this adaptation:
>
> |dataset: QH9-Stable-size-ood| unit (1e-6 Ha)
> |QH-net(reported) | QH-net(reproduced) | LEM(Ours) |
> |75.0 | 83.2 | **57.7** |
>
> More formal results will be added to the appendix of the manuscript after the conclusive test shortly.
> Therefore, rather than "breaking down," our method maintains validity within the DFT framework across different physical regimes, with adjustable parameters to accommodate varying degrees of locality.
>
> ***Q2. It is not clear from the paper (at least I could not find it) which is the cutoff used, or if different cutoffs were used for different systems. This could enable comparing the receptive field of the model proposed in this paper and other approaches using message-passing.***
>
> We apologize for not clearly specifying the cutoff selection. We will clarify this here, and add a description in the experiment section, in section 4.1 of the revised manuscript.
>
> Typically, we set the cutoff radius being the sum of two atomic orbital cutoffs used in DFT calculations for training labels. Specifically, for numerical atomic orbitals (used in SIESTA, ABACUS, OpenMX), The cutoff is explicitly defined in orbital files or generation inputs, typically 5-10 a.u. For Gaussian-type orbitals, The cutoff is determined during data processing as the distance beyond which matrix elements become negligible.
>
> The reason for this setting is that in a quantum operator matrix learning task, the cutoff radius needs to cover all significant matrix elements on the LCAO basis. For matrix elements between two atomic orbitals, their magnitude becomes negligible when the interatomic distance exceeds the sum of the two orbital cutoff radii. Therefore, our graph construction naturally aligns with this physical constraint by connecting atoms within this cutoff distance. A key advantage of our strict localized scheme is that the effective cutoff remains constant at this physical value, unlike MPNN-based methods (e.g., DeePH-E3, HamGNN) where it increases by a factor of L (number of network layers) due to message passing.

---

> > ### Author Response · Authors · 2024-11-22
> >
> > ***Q3:Evaluating the model performance in a transferability task would enhance significantly the submission.***
> > Thank you for this important suggestion about transferability evaluation. In our manuscript, we have demonstrated the model's transferability across different configurations from molecular dynamics trajectories, including varying atomic positions and cell sizes.
> > We would like to emphasize that our current tests focus on transferability across different configurations from molecular dynamics trajectories, including varying atomic positions and cell sizes. This type of transferability is of significant practical importance. In real applications, our SLEM model only requires training data from a very small portion of the MD trajectory and can effectively predict quantum operators over much longer time scales. This capability is particularly valuable when combined with Machine Learning Force Field (MLFF) methods, as it enables efficient computation of electronic properties during long-time simulations while minimizing the need for expensive DFT calculations. Many research scenarios focus on studying the dynamic evolution and properties of specific materials or molecular systems, where the primary concern is precisely this type of configuration-based generalization, rather than cross-chemical transferability.
> > Additionally, we have also studied the transferability beyond single-material configurations, we have now included tests on the QH9 dataset, which involves transfer learning across different molecular structures and varying numbers of atoms. However, since the screening effect is comparably weak in molecular systems, our test result in the QH9 dataset is performed with the LEM model, a variant of SLEM that has exponential decay, but trainable radial dependency. The QH9 results show:
> >
> > |dataset: QH9-Stable-size-ood| unit (1e-6 Ha)
> > |QH-net(reported) | QH-net(reproduced) | LEM(Ours) |
> > |75.0 | 83.2 | **57.7** |
> >
> > More detailed results will be updated in the manuscript shortly. We would like to further test our model's capacity in cross-system transferability of periodic materials. While it is not available yet due to the lack of an open-source dataset to support it, we would be happy to design a systematic data generation procedure, performing the test while contributing to the communities.
> >
> > Regarding cross-element transferability, this is indeed a more challenging task. While our model architecture theoretically supports multiple elements through its atom-type encoding and equivariant design, achieving reliable cross-element transferability typically requires a large-scale model trained on extensive data across many elements. This would involve comprehensive datasets covering diverse chemical environments, and more sophisticated training strategies to balance various element combinations. While this is an important direction for the field, it falls outside the scope of our current work, which focuses on demonstrating the benefits of strict locality in equivariant models. We believe establishing the effectiveness of strict locality is a crucial first step - once validated, these principles could be incorporated into future larger-scale models targeting cross-element transferability.
> >
> > ***Q4. Evaluating the scaling of the model in a multi-GPU setting would also enhance significantly the submission.***
> >
> > The multi-GPU inference capability for large structures is indeed a crucial aspect that could significantly enhance our work. While our strict locality design theoretically enables efficient parallelization, similar to Allegro, the implementation is more challenging in our case. Unlike Allegro, which only deals with node features for force field training, our method requires handling edge features for quantum operator matrix predictions. This introduces additional complexity in graph partitioning and cross-GPU communication, particularly for example, the edge feature handling across partition boundaries, the load balancing considering both node and edge computations and the efficient communication patterns for matrix element construction, etc. This makes our parallelization procedure more complex compared to Allegro's approach. It is currently an ongoing work requiring substantial development and validation. While we couldn't include comprehensive multi-GPU results in this manuscript, we believe our strict localization design is an essential step toward enabling DFT-level electronic structure calculations for device-scale systems. We are actively working on addressing these technical challenges.
> >
> > **References**
> >
> > [1] Resta, Raffaele. "Thomas-Fermi dielectric screening in semiconductors." Physical Review B 16.6 (1977): 2717.
> >
> > [2] Ninno, Domenico, et al. "Thomas-Fermi model of electronic screening in semiconductor nanocrystals." Europhysics letters 74.3 (2006): 519.

---

> > > ### Author Response · Authors · 2024-11-22
> > >
> > > In extra, we also want to highlight the creative invariant parameterization of the overlap matrix, as one of our most novel contributions. We fully utilized the two-centre nature of the Overlap Matrix and reduced the complex equivariant features into two-centre dependent scalar variables. Therefore, it can be directly predicted from the initialized two-centre embedding in the first layer of our network with minimal additional parameters. As described in the last paragraph of the introduction and method 3.2 section. In section 4.1, we show that fitting overlap along with the Hamiltonian/density matrix only increases 0.01M(~1.4%) parameters of the network. This is almost free for the network training, while it benefits vastly in model inference since the trained model with overlap can be independent of DFT software to compute the desired physical properties directly. We also add a sentence in the abstract to avoid mismatch.

---

> > > > ### Comment · Reviewer_nQRm · 2024-11-25
> > > > **Reply to authors**
> > > >
> > > > I would like to the thank the authors for the effort they have put in making extensive clarifications to all reviewers. Regarding my specific concerns, I am satisfied with all their replies. I will justify raising my score:
> > > >
> > > > 1) Authors have motivated very well their assumptions on locality (also to other reviewers), while acknowleding and identifying the systems were this assumption would break down. This motivated the modification of the architecture (SLEM to LEM) to handle these other cases, using the QH9 as a benchmark. I would encourage the authors to try to explain, in case of acceptance, the reasonings exposed here during the rebuttal, since I believe they improve the manuscript.
> > > > 2) Following from using QH9 as a benchmark, authors have addressed chemical transferability.
> > > > 3) The availability of new training sets authors have built is positively valued. Some more extended description would be necessary in the manuscript / supplementary material, and the authors have already mentioned that they plan to include it.
> > > > 4) My main research expertise is on MLIPs rather than direct quantum operator prediction, but familiary with previous literature in the field of equivariant graph neural networks and e3nn** in general has been enough to grasp the idea. Of course, the topic is complex, and perhaps it could be pedagogically introduced in some sort of supplementary material, but I don't feel the authors' exposition is particularly challenging to follow, quite the contrary. If the authors manage to use the extra space in case of acceptance for the introduction of some of the justifications they have made here, even better.
> > > > 5) I personally understand that deployment on multiGPU is a challenge on its own. I think that the authors have set the correct framework to work towards this implementation (at least for the systems where the locality assumption holds).
> > > >
> > > > I am strongly of the opinion that this manuscript should be accepted.
> > > >
> > > > **As the authors say, e3nn is a general purpose library that allows one to build architectures on top of it, not a specific architecture per se. Therefore, reporting MAEs using e3nn doesn't make sense.

---

> > > > > ### Author Response · Authors · 2024-11-26
> > > > >
> > > > > Hi, we sincerely appreciate your kind words. We are delighted you like this work and feel encouraged by seeing our answers address the question, and are also thankful for your kindness in helping us defend some of the hard critics.
> > > > >
> > > > > We agree with your suggestion to include the discussion with reviewers in the paper. We are working to contain the topic, such as the theoretical justification of the locality approximation of SLEM, and the details of the experiment on QH9 in the paper. We updated a milestone version and will update the final version shortly.

---

### Official Review · Reviewer_VuZs · 2024-10-31

**Soundness:** 3
**Presentation:** 3
**Contribution:** 3
**Rating:** 6
**Confidence:** 5

**Summary:**

The paper introduces Strictly Local Equivariant Message-passing (SLEM), a deep learning model designed to predict multiple quantum operator matrices, such as Hamiltonians, density matrices, and overlap matrices, within the density functional theory (DFT) framework. SLEM tries to address key challenges in efficiency and scalability that traditional methods face when handling large quantum systems. The authors validate SLEM’s performance across 2D and 3D material systems, demonstrating high accuracy for the experiments.

**Strengths:**

By focusing on a strictly localized equivariant message-passing framework, the authors present a creative way to address the challenges of efficiency and scalability in quantum mechanical computations. The use of SO(2) convolutions to manage high-order tensor complexity is particularly novel, as it reduces the computational burden associated with f and g orbitals.

The methodological rigor of the paper is detailed theoretical justifications for the design choices of SLEM. The authors provide mathematical foundations for the model's ability to preserve physical symmetries while maintaining strict locality utilizing quantum mechanical properties. The paper is well-organized, with a logical flow from the problem motivation to the model formulation and experimental validation.

The contributions of this work is important, especially for the fields of quantum chemistry and materials science. The ability to efficiently predict quantum operators without expanding the receptive field could accelerate DFT calculations and open new possibilities for simulating large-scale quantum systems.

**Weaknesses:**

The experimental comparisons presented in the paper are limited to only two other models, and these comparisons are not consistently provided for all experiments. Expanding the range of baseline models, including more well-established methods, would strengthen the validation of SLEM’s computational efficiency, scalability, and accuracy. Incorporating additional well-known benchmark datasets, such as QH9 [1], nablaDFT [2], and potentially QM9 [3, 4] (used in models like HamGNN), could provide a more reliable and widely recognized basis for evaluation. This would help clarify the practical advantages of SLEM more comprehensively. Additionally, while the authors conduct in-house simulations for some datasets, details about these datasets are not provided. More transparency about the data generation process and the choice of neural network potentials for sampling MD simulations would address concerns about the accuracy, quality, and reproducibility of the dataset. Given that datasets are often created using well-established computational methods like DFT, clarifying these choices would be beneficial.

Moreover, the mathematical formulation of SLEM is quite complex and may be difficult for practitioners who are not experts in quantum mechanics or advanced tensor operations. Providing a more accessible explanation or simplified overview could make the approach more approachable. Additionally, combining some of the experimental tables into one could improve readability and streamline the presentation.

In addition, the mathematical formulation of SLEM is highly complicated and could be difficult to implement for practitioners not deeply familiar with quantum mechanics and advanced tensor operations. The paper could be improved by providing a more accessible explanation. Furthermore, some of the experiment tables can be merged into one which would improve the readibility.


[1]: Yu, H., Liu, M., Luo, Y., Strasser, A., Qian, X., Qian, X., & Ji, S. (2024). Qh9: A quantum hamiltonian prediction benchmark for qm9 molecules. Advances in Neural Information Processing Systems, 36.

[2]: Khrabrov, K., Shenbin, I., Ryabov, A., Tsypin, A., Telepov, A., Alekseev, A., ... & Kadurin, A. (2022). nabladft: Large-scale conformational energy and hamiltonian prediction benchmark and dataset. Physical Chemistry Chemical Physics, 24(42), 25853-25863.

[3]: Ruddigkeit, L., Van Deursen, R., Blum, L. C., & Reymond, J. L. (2012). Enumeration of 166 billion organic small molecules in the chemical universe database GDB-17. Journal of chemical information and modeling, 52(11), 2864-2875.

[4]: Ramakrishnan, R., Dral, P. O., Rupp, M., & Von Lilienfeld, O. A. (2014). Quantum chemistry structures and properties of 134 kilo molecules. Scientific data, 1(1), 1-7.

**Questions:**

1) What was the reason for selecting only two models for comparison with SLEM, and how were these models chosen? Would you consider expanding your evaluation to include additional, widely recognized benchmark datasets, as previously suggested?

2) In the computational scalability analysis, SLEM was compared with E3NN. However, I noticed that E3NN’s performance metrics, such as MAE, were not included in the accuracy tables. Did E3NN achieve better accuracy compared to SLEM, even if it was more computationally expensive?

3) The paper states that structures for the Si, GaN, and HfO₂ systems were sampled via molecular dynamics using neural network potentials. Since these potentials may introduce approximations relative to traditional quantum mechanical methods like DFT, could you clarify the potential impact on the quality and reliability of your training data? How do you ensure that these approximations do not compromise the accuracy and generalizability of SLEM’s predictions?

4) Could the authors share more details about the in-house simulations used for data generation, such as specific parameters and configurations? Providing this information would greatly aid in the reproducibility of the study as well as the quality of the data.

---

> ### Author Response · Authors · 2024-11-19
>
> We thank the reviewer for the suggestions and questions. Some suggestions improve reliability and provide direct guidance for improvement. For example, we fully understand your concerns about the soundness of the experiment, data generation processes, and the comparison of open benchmarks. To address these concerns, we added detailed data generation processes and uploaded all our datasets for reproducibility.
>
> On the other hand, we want to highlight that quantum operator learning for periodic material systems is still an emerging field without standardized open benchmarks. We've done our best to increase the soundness of our work by using data from multiple sources and comparing it with the available methods. Below, we will carefully address the comments point by point and respectfully request the reviewer reconsider the marks, considering the novelty and contribution of this work.
>
> ***Q1.	What was the reason for selecting only two models for comparison with SLEM, and how were these models chosen? Would you consider expanding your evaluation to include additional, widely recognized benchmark datasets, as previously suggested?***
>
> Response: For the model selection, we use DeepH-E3 and HamGNN since they are the most representative (if not the only available) equivariant models in this area, that support Hamiltonian prediction task on periodic materials. While there exist other models that also predict the Hamiltonians, they focus on the molecular system and do not support periodic materials that our SLEM model specialises for. Therefore, our comparison with these two models represents the most comprehensive evaluation possible within the current landscape of the field.
>
> For the dataset selection, we share the reviewer's interest in evaluating recognized open benchmark datasets. However, unlike in molecules, widely recognised benchmark datasets such as QH-9 and nabla-DFT for Hamiltonian/Density/Overlap operators on periodic materials are unavailable until today. Therefore, to ensure comprehensive evaluation, we used a combination of public and in-house datasets:
> - For 2D materials: We utilized the public graphene and MoS2 datasets from DeepH-E3 model [X. Gong, et. al, Nat Commun 14, 2848 (2023)]
> - For 3D materials: Due to the lack of public datasets, we generated our data using ABACUS DFT software [P. Li, et. al, Comput. Mater. Sci. 112, 503 (2016)], which is openly available and uploaded on https://www.aissquare.com/datasets/detail?pageType=datasets&name=Quantum_Operator_Dataset&id=286.
>
> Our model demonstrated a consistent advantage in accuracy in all the above-mentioned datasets. We believe this proves the advance of our method.
> While SLEM primarily targets periodic systems, as suggested by the reviewer, we also evaluated its performance on molecular datasets like QH-9. For these smaller systems where all atom pairs fall within typical cutoffs (resulting in fully connected Hamiltonians), we implemented a semi-local variant called LEM (Localized Equivariant Message-passing). Our comparative analysis with QH-net on the QH-9 benchmark (will be updated to the appendix shortly) demonstrates excellent accuracy, further validating our method's versatility across both periodic and molecular systems.
>
>
> ***Q2. In the computational scalability analysis, SLEM was compared with E3NN. However, I noticed that E3NN's performance metrics, such as MAE, were not included in the accuracy tables. Did E3NN achieve better accuracy compared to SLEM, even if it was more computationally expensive?***
>
> Response: We would like to clarify that E3NN's performance metrics (MAE) are not included in our accuracy comparison tables because E3NN is a general-purpose equivariant neural network library rather than a specific model for quantum operator prediction. E3NN does not provide ready-to-use implementations for quantum operator learning tasks, making a direct accuracy comparison impossible.
>
> The E3NN comparison in our computational scalability analysis serves a different purpose - comparing tensor-product implementations. The standard tensor product implementation in the E3NN package, while widely used, becomes computationally intensive when handling materials with higher orbitals. DeepH-E3 improved upon this by using a low-rank approximation for the weight matrix, and our SLEM model further advances this direction through an SO(2)-based tensor product approach. This comparison specifically examines the computational efficiency of these different tensor product operations, rather than model accuracy. We sincerely hope our clarification satisfies your question.

---

> > ### Author Response · Authors · 2024-11-19
> >
> > ***Q3.	The paper states that structures for the Si, GaN, and HfO2 systems were sampled via molecular dynamics using neural network potentials. Since these potentials may introduce approximations relative to traditional quantum mechanical methods like DFT, could you clarify the potential impact on the quality and reliability of your training data? How do you ensure that these approximations do not compromise the accuracy and generalizability of SLEM's predictions?***
> >
> > Response: For short answer, using an approximated force field does not affect the data quality of the quantum operator learning task, since the structure sampling and the quantum operator labelling process are fully independent.
> >
> > We want to emphasize that the molecular dynamic with neural network potentials is only used to sample and generate diverse atomic configurations (spatial coordinates) of the target systems (here the the Si, GaN, and HfO2). For each sampled structure, independent DFT calculations are performed to obtain the quantum operators as training labels. Since the usage of a force field does not affect the accuracy of the data label, therefore the mapping from structure to the quantum operators learned by our model would not compromise the accuracy and generalizability of the sampling method. We hope this could address the potential concerns.
> >
> > ***Q4.	Could the authors share more details about the in-house simulations used for data generation, such as specific parameters and configurations? Providing this information would greatly aid in the reproducibility of the study as well as the quality of the data.***
> >
> > Response: We highly agree with the reviewer's option about data openness. We are very willing to share all the details of our data generation process, and configurations to help improve the reproducibility. Therefore, we upload all the training data and configuration of this work along with a usage instruction notebook in the dataset opensource website AISquare with the link:  https://www.aissquare.com/datasets/detail?pageType=datasets&name=Quantum_Operator_Dataset&id=286 .
> > We also added a new "DATA GENERATION" section in the appendix to discuss the detailed generation process, packages and parameters used to generate our data. We hope these effects will aid in the reproducibility and quantity of our study.
> >
> > Additionally, we also want to highlight the creative invariant parameterization of the overlap matrix, as one of our most novel contributions. We fully utilized the two-centre nature of the Overlap Matrix and reduced the complex equivariant features into two-centre dependent scalar variables. Therefore, it can be directly predicted from the initialized two-centre embedding in the first layer of our network with minimal additional parameters. As described in the last paragraph of the introduction and method 3.2 section. In section 4.1, we show that fitting overlap along with the Hamiltonian/density matrix only increases 0.01M(~1.4%) parameters of the network. This is almost free for the network training, while it benefits vastly in model inference since the trained model with overlap can be independent of DFT software to compute the desired physical properties directly. We also add a sentence in the abstract to avoid any omission.

---

> ### Comment · Reviewer_VuZs · 2024-11-25
>
> I would like to thank the authors for their thoughtful responses and the progress highlighted in their rebuttal. I have updated my ratings for the paper.

---

> > ### Author Response · Authors · 2024-11-26
> >
> > Hi, we appreciate that our response and progress addressed your questions. We are encouraged by the updated review and scores and feel very happy that you enjoyed this paper.

---

### Official Review · Reviewer_hW5o · 2024-11-02

**Soundness:** 4
**Presentation:** 3
**Contribution:** 3
**Rating:** 8
**Confidence:** 3

**Summary:**

This paper proposes a data driven method to solve KS-DFT. Instead of solving the KS system to consistency, this method puts the configuration of atoms through a carefully designed SO(2) neural network and directly predicts the quantum operators at self consistency. While there are many other existing works trying to accomplish the same thing, the key technical contributions of this paper can be summarized as the following two points:

- The SLEM architecture, compared to traditional methods, SLEM has a strictly local design, its effective cutoff does not increase as more layers are added. This architecture is more scalable/parallelizable because the dependency is much smaller.
- The parameterization of the invariant overlap operators, which enables the prediction of overlap operator without requiring lots of parameters.

The above innovations is verified to be effective in the empirical evaluations on various systems. The advantages reported include

- Better generalization due to the more restricted model assumption.
- Better scaling behavior w.r.t angular momentum.
- Faster iteration speed and smaller memory footprint.

**Strengths:**

- The two key contributions are novel, they are clearly explained in the paper and based on my understanding they are technically sound.
- The strictly local structure is significant and likely to be widely adopted in the future given the nice properties, not only more parallelizable, but also leads to lower errors.

**Weaknesses:**

I have a major concern on the way the evaluation is carried out

- The training and testing happens on the same system using trajectories of molecular dynamics. Although this type of evaluation may be also used in the baselines that the author compare to, I feel it is not sufficient. A good generalization is not surprising if the MD trajectories have a good coverage of different atomic geometric configurations. We run into the chicken egg dilemma, if DFT is already calculated on a system, why would we need to fit a blackbox model and do it again faster. In my opinion, the evaluation should be performed across different materials in a combinatoric way, i.e. train on material made of `AB` and `BC` and `AC`, evaluate on material made of `ABC`. Or on materials that contain same elements but in different proportions, i.e. train on `1A2B` and evaluate on `2A1B`.

Another concern is on the theoretical soundness

- Based on my understanding, only atoms pairs the has a distance smaller than rcut is considered, which means that the interaction between atomic orbitals from distant atoms are not considered, however, the operator needs to include all pairs of interactions to form a matrix. How are the noninteracting entries of the matrix set?
- Although the empirical study favors a strictly local structure, it is unintuitive theoretically. For example, the hartree term in DFT is a slow decaying term, using `(ij|kl)` to represent the four center integral, and `(i|j)` as the overlap; When `(i|j)` and `(k|l)` are both large, the `(ij|kl)` term is not negligible even when `ij` is distant from `kl`. It would added to the soundness of this paper if the authors could provide a theoretical support.
- With the above said, it is crucial to discuss the limitation of this method, i.e. in which scenario would this method fail due to the strictly local assumption.

**Questions:**

My questions are stated in the above concerns, I would be happy to raise my score if the authors address them.

---

> ### Author Response · Authors · 2024-11-16
>
> **Dear reviewer:**
>
> We sincerely thank you for your careful review and constructive feedback on our paper. We particularly appreciate your recognition of our two key technical contributions - the strictly local SLEM architecture and the parameterization of invariant overlap operators. Your positive assessment of these innovations as being "novel" and "technically sound" is very encouraging.
>
> In your review, you raised two main concerns: one regarding our evaluation testing along the trajectories of molecular dynamics, and another about the theoretical soundness about the strict local design of our approach, especially for the Hartree term in DFT. We would like to address these concerns point by point:
>
> **1.  For the first concerns about the generalization test on MD trajectory, and suggestion of test on transferring to different materials with same chemical species.**
>
> We appreciate the reviewer's suggestion about cross-system generalization tests. While such extensive transferability (training on AB, BC, AC systems and testing on ABC) would be valuable for general-purpose models, we want to point out that specialized models trained for specific systems are both sufficient and preferable in many practical applications. Many research scenarios focus on studying the dynamic evolution and properties of specific materials or molecular systems, where the primary concern is the model's ability to generalize across different configurations of the same system, rather than across entirely different chemical compositions.
> Nevertheless, to demonstrate our model's capability for cross-system generalization, we have conducted evaluations on the well-established QH9 dataset, where the training and testing sets consist of molecules with different chemical compositions and varying numbers of atoms. Our model has shown excellent transferability in these tests, achieving competitive performance compared to the reported results of QH-net. We list the current results here, and more extensive results will be updated in the paper shortly.
>
> |dataset: QH9-Stable-size-ood| unit (1e-6 Ha)
>
> |QH-net(reported) | QH-net(reproduced) | LEM(Ours) |
>
> |75.0                     | 83.2                          | **57.7**  |
>
> Additionally, we would like to emphasize that the generalization capability along MD trajectories is of significant practical importance. In real applications, our SLEM model only requires training data from a small portion of the MD trajectory. Once trained, it can effectively predict quantum operators over much longer time scales. This capability is particularly valuable when combined with Machine Learning Force Field (MLFF) methods. In such scenarios, DFT calculations are only needed for a short time period to generate the energy, force, Hamiltonian/overlap, and density matrix data for training both MLFF and SLEM models. Subsequently, by running AIMD with MLFF and predicting the Hamiltonian along long-time simulations, we can efficiently compute many important electronic properties that are time/temperature dependent or only emerge after extended MD simulations. This approach significantly reduces the computational cost of DFT calculations while maintaining accuracy.

---

> ### Author Response · Authors · 2024-11-16
>
> **2. For the second concern about the theoretical soundness of the strict local design:**
> The theoretical foundation for our local design comes from the behaviour of Hamiltonian matrix elements. For example, the Hartree potential matrix element between two atomic orbitals takes the form:
> $$V_{ij}^H=\int\phi_i^*(r-R_A)V_H(r)\phi_j(r-R_B)dr$$
> where $R_A$  $R_B$  are the centres of atomic orbitals, this integral decays rapidly with the distance $R_A-R_B$ due to the localized nature of atomic orbitals.
> The reviewer raised an important point about the four-center integral (ij|kl) which appears in the Hartree-Fock method or the hybrid functionals DFT:
> $$(ij│kl)=\int\int\phi_i^*(r_1-R_A )\phi_j(r_1-R_B )\frac{1}{|r_1-r_2|}\phi_k^*(r_2-R_C)\phi_l(r_2-R_D )dr_1dr_2$$
> Indeed, this integral can be significant even when the ij pair is distant from the kl pair, because when both overlaps (i|j) and (k|l) are large (meaning $|R_A-R_B|$ and $|R_C-R_D|$ are small), the Coulomb interaction $1/|r1-r2|$ decays slowly with the distance between the pairs. However, our work primarily focuses on standard DFT with local/semi-local exchange-correlation functionals (LDA/GGA/meta-GGA), where such four-center integrals do not appear in the Hamiltonian. As for the hybrid functionals, though not the focus of our work, due to the screening effects, the Coulomb interaction decays more rapidly, and we can still treat it using the local framework by using a larger cutoff, as has been demonstrated in recent work in DeepH [Tang, Z. et al. Nat. Commun. 15, 8815 (2024)].
>
> In our implementation, the cutoff radii (rcut) are chosen based on the inherent localization of atomic orbitals used in DFT calculations. Since typical DFT orbital cutoffs range from 5 to 10 Bohr, our model's cutoff for the distance between orbital centres $(R_A-R_B)$ is naturally set to twice this value (10-20 Bohr).
> For periodic systems, this cutoff is quite natural - matrix elements for orbital pairs beyond this distance are set to zero, which is consistent with the locality of the basis functions. For small molecules, where all atoms might fall within this cutoff range, all orbital pairs are considered, effectively becoming a full-range calculation. This explains why our model performs well across different types of systems while maintaining its physically-motivated local design.
> Besides, to further improve the flexibility and applicability of our method, we also developed a semi-localized version called LEM, where the interaction between distant atoms is included but decays exponentially with their distance, with a trainable decay factor.  We agree with the reviewer's concern and add a section to discuss the limitations of this method.

---

> ### Comment · Reviewer_hW5o · 2024-11-18
>
> Thanks for the updates on the empirical evaluation.
> For the strict locality, I'm not entirely aligned yet. SLEM not only performs well, but actually outperforms is still quite anti-intuitive to me.
>
> > our work primarily focuses on standard DFT with local/semi-local exchange-correlation
>
> I was not talking about the exact exchange in hartree fock, but on the computation of coulomb interaction (hartree term) which is also computed via contracting with the four center integral. As you agreed above, distant shell pairs would still have an non-negligible four center integral, and consequently non-negligible in the hartree term. The Cauchy-Schwartz inequality used in integral screening does help remove some of the entries, but I doubt using a hard cutoff is legit, otherwise it would have already been applied in electron integral.
>
> > Since typical DFT orbital cutoffs range from 5 to 10 Bohr
>
> Not sure what you mean by "orbital cutoff range". I didn't find a description on what type of orbitals are used in this study, then I assume it is GTOs being used, which would not have any hard cutoff (thus the above discussion on the four center integrals). If you're using strictly local numerical orbitals then these would make sense.
>
> > For periodic systems, this cutoff is quite natural
>
> In periodic systems, the computation often breaks down into a local and a long range term. The long range effects still need to be considered, and should vary with the configuration.
>
> > LEM
>
> Good to know this variant, could you comment on the relative performance of LEM and SLEM?

---

> ### Author Response · Authors · 2024-11-22
>
> **For the anti-intuition of outperforming SLEM**
>
> We fully understand your concern. The SLEM model introduces extra constraints and, therefore should not be more accurate than well well-constructed MPNN model. We agree with this consideration. The good accuracy of SLEM, from our understanding, should be contributed mostly by the model design, parameterization of the equivariant features (as in 3.1, we standardized all the irreducible matrix element features using statistical quantities from the dataset) and the usage of SO(2) convolution. However, SLEM's data efficiency in periodic material systems mostly comes from the strict locality, as it helps the model to capture the dependency more physically in localized systems with a strong physical prior.
>
> **About the long-range effect of Hartree Term**
>
> $$
> V_{ij}^H=\int\phi_i^*(r-R_A)V_H(r)\phi_j(r-R_B)dr
> $$
> $$
> =\int\phi_i^*(r-R_A)\left[\frac{1}{4\pi\epsilon_0}\int\frac{\rho(r')}{|\vec{r}-r'|}dr'\right]\phi_j(r-R_B)dr
> $$
> $$
> =\int\phi_i^*(r-R_A)\left[\sum_{kl}\frac{1}{4\pi\epsilon_0}\int\frac{\phi_k^*(r'-R_C)\phi_l(r'-R_D)}{|\vec{r}-r'|}dr'\right]\phi_j(r-R_B)dr
> $$
> $$
> =\frac{1}{4\pi\epsilon_0}\sum_{kl}\int\int\frac{\phi_i^*(r-R_A)\phi_k^*(r'-R_C)\phi_l(r'-R_D)\phi_j(r-R_B)}{|\vec{r}-r'|}dr'dr
> $$
>
>
> Indeed, if we split the Hartree term in the above form, there exist four centre integrals that $\langle ik|lj\rangle$ is large even when A and B are far away from each other. However, the integration results are a collective effect that sums over all pairs of $kl$, which is hard to interpret directly. Moreover, we are looking for a locality in the Hamiltonian matrix element. It is a combination of the LCAO basis, hartree term, external potential, and xc functional. Therefore, we need to discuss the localization as a collective effect, which is contributed by the following properties:
>
> $$V_{ij}^H=\int\phi_i^*(r-R_A)\left[V_H(r)+V_{ext}(r)+V_{xc}(r)\right]\phi_j(r-R_B)dr$$
>
> 1. **The decaying behaviour of the LCAO basis.**
>
>     For all LCAO bases (whether GTO, NAO, or Slater-like), the radial part all decays rapidly with distance. Therefore, the long-range term in $V_H, V_{ext}, V_{xc}$'s contribution in the integration (that decays as 1/r) would be decreased fastly (often exponentially such as in GTO or Slater-like orbital).
>
> 2. **The system's electronic neutralization condition.**
>
>     The electronic neutralization condition requires that the long-term part of the Hartree term and the external term cancel each other. To derive this, we assign the electron density to the atomic center, by writing as:
>     $$n(r)=\sum_In_I(r-R_I)$$
>     Then the sum of $V_H(r), V_{ext}(r)$ would be:
>
>     $$-\sum_I\frac{Z_I}{|r-R_I|}+\sum_I\int\frac{n_I(r'-R_I)}{|r-r'|}dr'$$
>
>     When $|r-R_I|$ is large, we can approximate $|r-r'|\approx|r-R_I|$, therefore:
>     $$-\frac{Z_I}{r-R_I}+\int\frac{n_I(r'-R_I)}{|r-r'|}dr'\approx-\frac{Z_I}{|r-R_I|}+\frac{Z_I}{|r-R_I|}=0$$
>
> 3. **The screening effect.**
>
>     we refer to [1,2] the study of screening radius $R_s$, which describes the system's electrostatic potential's reaction to the vibration of charges.
>
>     In [1], the screening radius of typical insulator and semiconductor systems is reported as:
>
>     Diamond: 2.76 a.u.
>
>     Silicon: 4.28 a.u.
>
>     Germanium: 4.71 a.u.
>
>     In [2], for some nanoparticles, the $R_s$ is reported as:
>
>     Si191H148: 5.36 a.u.,
>     Ge191H148: 5.61 a.u.
>
> Combining with the above effects, we can see the $H_{ij}$ Hamiltonian elements under LCAO basis can be approximated locally dependent on the neighbouring area of atom $A,B$.

---

> ### Author Response · Authors · 2024-11-22
>
> **About the DFT orbital cutoff**
>
> We are using Numerical Atomic Orbitals (NAO) in this study. Generally, numerical atomic orbital has a cutoff radius, beyond which, the orbital value is ignored since it is too small. This type of basis is widely used in LCAO DFT software for periodic systems such as SIESTA, OpenMX, ABACUS etc. For molecule systems, the GTO are more generally accepted and Indeed does not have a certain cutoff radius. However, since the basis decayed exponentially with radial distance, the cutoff is defined as the value beyond which the quantum operator matrix elements are small to ignore, which can be obtained during the data processing step.
>
> We apologise for the lack of a detailed description of DFT calculations. We employ ABACUS[3] software to generate our data, and use NAO basis set generating follows [4,5] using SG15 pesuodopotential [6]. We have added a section in the appendix to discuss the data generation process, and a section for the cutoff selection criteria with more detail.
>
> **About LEM**
>
> The accuracy of the LEM model on some periodic systems reported in the paper (such as silicon) is similar to SLEM, while the data efficiency of LEM is not as good. However, in other cases when the screening effect is weak (for example the metal-molecule junction we studied recently), the SLEM would be less accurate than LEM. We need to admit that we only have limited comparison examples about SLEM and LEM since this paper is mostly focused on the prior one. However, we highly agree that a thorough comparison of local, semi-local and conventional MPNN models to discover the specific applied area would be a great contribution both for theoretical understanding and to guide the user communities. We are looking forward to studying this in future work.
>
>
> [1] Resta, Raffaele. "Thomas-Fermi dielectric screening in semiconductors." Physical Review B 16.6 (1977): 2717.
>
> [2] Ninno, Domenico, et al. "Thomas-Fermi model of electronic screening in semiconductor nanocrystals." Europhysics letters 74.3 (2006): 519.
>
> [3] Li, Pengfei, et al. "Large-scale ab initio simulations based on systematically improvable atomic basis." Computational Materials Science 112 (2016): 503-517.
>
> [4] Peize Lin, Xinguo Ren and Lixin He, Strategy for constructing compact numerical atomic orbital basis sets by incorporating the gradients of reference wavefunctions, Phys. Rev. B 103, 235131 (2021).
>
> [5] Mohan Chen, Guang-Can Guo, and Lixin He, Systematically improvable optimized atomic sets for ab initio calculations, J. Phys.: Condens. Matter 22, 445501 (2010).
>
> [6] Hamann, D. R. "Optimized norm-conserving Vanderbilt pseudopotentials." Physical Review B-Condensed Matter and Materials Physics 88.8 (2013): 085117.

---

> > ### Comment · Reviewer_hW5o · 2024-11-23
> >
> > Thanks for the additional feedback.
> > I agree that in periodic systems the neutralisation effect would dominate and hence the eri with longer range could be less important. Using NAO with strict cutoff resolves my questions. Setting aside the restrictions of this cutoff in the original DFT calculation, it is legit to say that SLEM is capable of approximating it.

---

> > > ### Author Response · Authors · 2024-11-23
> > >
> > > Hi, thanks for the understanding and appreciation. We are very encouraged by your words and feel very happy to see the scores you raised. We hope the method employed in this paper can contribute to this field.

---

### Meta-Review · Area_Chair_bNh1 · 2024-12-18

**Metareview:**

The paper proposes an equivariant neural network that predicts Hamiltonian (converged Fock matrix in DFT) for molecular systems. The authors highlighted the benefit of reduced computational and memory cost by imposing a strictly local design to the architecture, together with other tricks e.g. a proper processing of the overlap matrix, and the usage of SO(2) convolution for lighter cost in tensor product.

I agree with reviewers' appreciation that this work contributes to the bottlenecking issue of efficiency in this domain, and even showing better generalization. Some concerns are raised by the reviewers, including the rationale of the strictly local assumption, generalization results over chemical space, choices of baselines and metrics and dataset details. In the rebuttal, the authors provided a reasonable argument for the locality at least for periodic systems, further results on QH9 that can be accounted for chemical space generalization evaluation, and explanations on baselines and metrics and enriches details in curating the dataset. Correspondingly, two reviewers have raised their scores.

In addition, I also asked two questions regarding the SOTA claim (considering e.g. PhiSNet) and the novelty of using SO(2) convolution (considering MLFFs that use this). The reply says that the proposed method mainly targets at periodic systems while PhiSNet at molecular systems, and that the proposed method outperforms QHNet on QH9 which has a similar performance as PhiSNet, and that the contributions of strictly local design and parameterization of the overlap matrix are more central contributions. This could be an acceptable answer, but I hope the authors could provide more details on the experimental setup and protocols to consolidate that the improvement is not from extensive tuning, and reasoning the improvement on QH9 given that the authors emphasized the beneficial cases are mainly periodic ones due to better satisfaction of quantum screening.

In respect to reviewers' preference, I recommend an accept to this submission. But I hope the authors could further improve the paper accordingly for the final version, e.g., the more comprehensive explanations on the locality assumption, their own dataset details, and configurations on the QH9 dataset, apple-to-apple comparison with other baselines, and interpretation of the better generalization in the non-periodic case.

**Additional Comments On Reviewer Discussion:**

I agree with reviewers' appreciation that this work contributes to the bottlenecking issue of efficiency in this domain, and even showing better generalization. Some concerns are raised by the reviewers, including the rationale of the strictly local assumption, generalization results over chemical space, choices of baselines and metrics and dataset details. In the rebuttal, the authors provided a reasonable argument for the locality at least for periodic systems, further results on QH9 that can be accounted for chemical space generalization evaluation, and explanations on baselines and metrics and enriches details in curating the dataset. Correspondingly, two reviewers have raised their scores.

In addition, I also asked two questions regarding the SOTA claim (considering e.g. PhiSNet) and the novelty of using SO(2) convolution (considering MLFFs that use this). The reply says that the proposed method mainly targets at periodic systems while PhiSNet at molecular systems, and that the proposed method outperforms QHNet on QH9 which has a similar performance as PhiSNet, and that the contributions of strictly local design and parameterization of the overlap matrix are more central contributions. This could be an acceptable answer, but I hope the authors could provide more details on the experimental setup and protocols to consolidate that the improvement is not from extensive tuning, and reasoning the improvement on QH9 given that the authors emphasized the beneficial cases are mainly periodic ones due to better satisfaction of quantum screening.

---

### Decision · Program_Chairs · 2025-01-22

Accept (Spotlight)